# Edible Insects in Africa in Terms of Food, Wildlife Resource, and Pest Management Legislation

**DOI:** 10.3390/foods9040502

**Published:** 2020-04-16

**Authors:** Nils Th. Grabowski, Séverin Tchibozo, Amir Abdulmawjood, Fatma Acheuk, Meriem M’Saad Guerfali, Waheed A.A. Sayed, Madeleine Plötz

**Affiliations:** 1Institute for Food Quality and Food Safety, Hannover University of Veterinary Medicine, Foundation, Bischofsholer Damm 15, D-30173 Hannover, Germany; Madeleine.Ploetz@tiho-hannover.de; 2Research Centre for Biodiversity Management, 04 B.P., Cotonou BJ-0385, Benin; tchisev@yahoo.fr; 3Institute for Food Quality and Food Safety, Research Center for Emerging Infections and Zoonoses (RIZ), Hannover University of Veterinary Medicine, Foundation, Bünteweg 2, D-30559 Hannover, Germany; amir.abdulmawjood@tiho-hannover.de; 4Laboratory of Valorization and Conservation of Biological Resources, University of M’Hamed Bougara of Boumerdes, Avenue de l’indépendance, Boumerdès DZ-35000, Algeria; fatma.acheuk@yahoo.fr; 5Laboratory of Biotechnology and Nuclear Technologies LR16CNSTN01, National Centre for Nuclear Sciences and Technology, Technopole de Sidi Thabet, Sidi Thabet T-2020, Tunisia; msaad_tn@yahoo.fr; 6Biological Application Department, Nuclear Research Centre, Atomic Energy Authority, Cairo ET-11787, Egypt; waheedel@hotmail.com

**Keywords:** entomophagy, food law, Africa, food hygiene, food policy

## Abstract

Entomophagy is an ancient and actually African tradition that has been receiving renewed attention since edible insects have been identified as one of the solutions to improve global nutrition. As any other foodstuff, insects should be regulated by the government to ensure product quality and consumer safety. The goal of the present paper was to assess the current legal status of edible insects in Africa. For that, corresponding authorities were contacted along with an extensive online search, relying mostly on the FAOLEX database. Except for Botswana, insects are not mentioned in national regulations, although the definitions for “foodstuff” allow their inclusion, i.e., general food law can also apply to insects. Contacted authorities tolerated entomophagy, even though no legal base existed. However, insects typically appear in laws pertaining the use of natural resources, making a permit necessary (in most cases). Pest management regulation can also refer to edible species, e.g., locusts or weevils. Farming is an option that should be assessed carefully. All this creates a complex, nation-specific situation regarding which insect may be used legally to what purpose. Recommendations for elements in future insect-related regulations from the food hygiene point of view are provided.

## 1. Introduction

### 1.1. Entomophagy on the African Continent

Some years ago, the FAO (Food and Agricultural Organization of the United Nations) recognized the potential of edible insects as one possibility to mitigate hunger and the effects of the climate change, and as a response to that, the discussion of establishing insect farms in traditionally entomophagous countries rather than increasing the extraction from the wild started [1].

Edible insects have been part of the human diet from the dawn of mankind on. However, food habits changed over the millennia, and while consuming insects was largely lost in Europe after the classical antiquity, the tradition lingered on in Africa. There are hundreds of insect species consumed in Africa as foodstuffs or as traditional medicine [1,2,3,4,5,6]. The awareness of the benefits of edible insects has also reached non-traditional sectors of the African population, and web-based information sites like LINCAOCNET (http://gbif.africamuseum.be/lincaocnet_dev/) provide searchable information on local species.

Insects are traded in a relatively small to medium level. The economic benefit varies with the species and is seldom accounted for, but one of the most significant ones seems to be the phane caterpillars of a saturniid emperor moth *Gonimbrasia belina* (ex “Imbrasia belina”), reaching a yearly trade value of more than $85 million in Southern Africa.

Like with any other foodstuff, the consumption of edible insects may lead to consumer risks, typically allergens, foodborne diseases, food spoilage agents, and contaminants [7]. Being so, the tradition has developed a set of dos and do nots to ensure food safety to a certain degree. However, as traditions develop over long periods of time and tend to become inflexible, some parts of it may not cover “modern” risks like environmental pollution, or even packaging [3]. In fact, the traditional handling of African insect-based products has become submitted to scientific research, and results show that even processed products may contain pathogens. By means of illustration (and far beyond completeness of data), Table 1 provides a look into the microbiology of fresh and processed products from three African insect species.

These results suggest that a stricter control of insect-based products is needed, moreover, if insect entrepreneurs and/or consumers lose this traditional knowledge. Implementing food legislation is a proven method to reduce food-related consumer risks.

### 1.2. The European Union as a Starting Point for Legal Considerations

In Europe and in terms of food legislation, a division can be made between EU member and non-member states, and within the EU, between EU law and national law. These food laws, along with a large number of related legal texts, represent the base of public health control of foodstuffs. It covers all the productions steps of the food chain from the primary production to the purchase of the product by the consumer. With the EU, community and national legislation was harmonized largely to pursue a maximum of congruency among laws and a minimum of features regulated twice (i.e., on EU and national level). The system has been established for the ordinary animal-derived foodstuffs and is in constant revision and improvement.

Europe is one of those geographical areas with no recent entomophagy tradition (despite some exceptions), and the discussion is moreover addressing the feasibility and the practical and legal framework to establish insect farms. This created a dilemma, because on one hand, entrepreneurs that wish to start an insect business depend on the certification by corresponding authorities. On the other hand, in many countries, insects are neither expressly allowed nor forbidden. So, there is no legal framework by which these authorities could certify this enterprise, even if there is good will to promote this development. Thus, authorities ask the entrepreneurs for more information on risk handling, etc., which in turn can only generate once the business is running. This creates a climate of legal uncertainty, which is perceived as obstacles by the insect business operators [8,9].

However, this is changing. In 2015, the EFSA (European Food Safety Authority) published the *Risk profile related to production and consumption of insects as food and feed*, coming to the overall conclusion that, when produced according to current law requirements, insects do not pose a major risk to consumers. However, knowledge on residues and contaminants was scarce, and EFFSA recommended more research. The second important EU publication is the amendment of the novel food regulation, i.e., *REG (EC) 2015/2283*. In it, insects are clearly classified as potential foodstuffs, but for each species and product, a separate authorization procedure must be followed, leading to the inclusion of the novel foodstuff in the so-called Union List. At present, there are several requests, which are being processed. Although this regulation does not contain specific requirements for insects as foodstuffs (e.g., primary production, processing, quality parameters, etc.), this regulation provides the base for a clearer, EU-wide regulatory frame for all sectors involved in insect production.

Finally, there is draft *Ref. Ares (2019) 382900—23/01/2019*, which proposes to add another Appendix A, specific for edible insects to the *REG (EC) 853/2004*. This regulation contains all regulatory issues about the production of animal-based foodstuffs along the production chain. The draft contains a definition of “insect” and proposals on the choice of allowed feedstuffs in insect farming. It has, however, not been ratified so far.

Until then, several nations issued national guidelines, which should be regarded as recommendations and interim solutions. However, there is a marked degree of heterogeneity in this process on the continent, ranging from utter rejection to a relatively developed legal framework to produce, sell, and monitor insect-based foodstuffs (IBF). In addition, not all European nations have issued official statements regarding edible insects so that for those countries, a grey zone is presumed. This current European situation is described in [7].

### 1.3. African Countries and Economic Communities

Africa, as all other continents, is a spectacular mixture of ethnic groups, languages (approximately 2000), religions, lifestyles, and cultures, subsuming into more than 1.3 billion humans inhabiting it. All African states are part of the African Union (AU). Within it, the African Economic Community (AEC) is the common organization seeking, ultimately, economic and monetary union of the member states, which vaguely corresponds to the EU. AU/AEC recognize several, so-called Regional Economic Communities (REC), some of which have subgroups, i.e.,
Arab Maghreb Union (UMA);Common Market for Eastern and Southern Africa (COMESA);Community of Sahel-Saharan States (CEN-SAD);East African Community (EAC);Economic Community of Central African States (ECCAS), with CEMAC (Economic and Monetary Community of Central Africa) as subgroup;Economic Community of West African States (ECOWAS), with UEMOA (West African Economic and Monetary Union) and WAMZ (West African Monetary Zone) as subgroups; the latter does not address specific food or agriculture issues and is therefore not considered here;Intergovernmental Authority on Development (IGAD);Southern African Development Community (SADC), with SACU (South African Customs Union) as a subgroup.

However, most subgroups are not recognized by AEC (CEMAC, SACU, and UEMOA), and there are other trade unions that also lack recognition by AEC, i.e., the Economic Community of Great Lakes Countries (CEPGL), Indian Ocean Commission (IOC), and Mano River Union (MRU).

Finally, there are trade blocks that include both African and non-African nations. One example is GAFTA, the Greater Arab Free Trade Area, which originates from the Arab League and includes most Arabic-speaking countries inside and outside Africa. IOC is an organization that links African island nations with France via this country’s oversea regions. Table 2 subsumes the affiliation of the African nations to these trading communities. It reveals a complex network of interactions between the different African nations.

Regardless of the official status at AU/AEC, these trade blocks have the goal to improve the trade among the member states, creating free trading zones. The trade also includes foods, and just like many European countries, African countries face the problem of trading foodstuffs with different quality standards. Another of the challenges these blocs have is the fact that many African countries belong to the more than one trading block, and the activity of these blocks varies. Some of these blocks have issued common trade rules that can be read in the internet, and some do not.

Besides international rules, national food law and other regulations also a play a very important role, particularly for foodstuffs produced and consumed inside the corresponding countries. The extent of this legislation varies strongly among them.

The aim of this contribution is to outline the current legal situation regarding edible insects in all African countries.

## 2. Material and Methods

For this research, all African countries were included (Table 2). Not entering a political dispute, states not recognized by the United Nations (typically “break-away states”) were included if five or more UN nations recognized them as independent. In this way, Galmudug, Khaatumo, Puntland, Republic of Azania, and the Republic of Somaliland were not considered in this survey, while the Sahrawi Arab Democratic Republic was. However, no data for the latter could be encountered, so no further mention of this country would be made.

Initially, an approach similar that made for the publication regarding the legal status in Europe [7,10] was intended, i.e., contacting the competent authorities (Table 3) personally by telephone and/or mail. However, the degree of response varied strongly among countries, with the best replies from the most French-speaking and Arabic-speaking countries. Others did not respond (even after contacting the corresponding embassies in Europe), even after several attempts, and so their homepages were searched for corresponding regulations. As this was not successful in some of them, the FAOLEX page (http://www.fao.org/faolex/country-profiles/en/) was consulted, trusting in having the most complete and updated database for FAO-related legislations. Eventually, all nations were checked by this mean. In some cases, additional information was found in secondary sources, e.g., Droit-Afrique.com, but there is a possible bias because these regulations may not be reflecting the actual situation of the given country. Data was gathered between the fall of 2018 and that of 2019.

Resources considered in this paper were publications from the AU and other trade blocks detailed in Table 2 and regulations that have been in force. It largely omits national policy papers due to space reasons and because policies may be implemented late, in a changed version, or even be abandoned, not meeting this contribution’s title. Bilateral agreements among African states or between African and non-African states (which are basically trade agreements) were also excluded in this analysis, as they are of minor concern for the African citizens.

The use of insects other than food was also not considered here.

Regarding which insect species are edible or not, the list of Jongema [11] was taken as reference. Within the text, law references are marked in italics (sometimes abbreviated) and presented before the ordinary reference list as an Appendix A. Although written differently in the various documents, the abbreviations for “number” were unified according to the corresponding typing conventions, i.e., “№” (English), “n^o^” (French and Portuguese), and “n.^o^” (Spanish). Regarding capitalization, the original version of the acts was retained.

## 3. Results

In Africa, edible insects may be regulated mainly in three different contexts:Food law;Wildlife resources management;Pest management.

The applicability of these regulations will depend fundamentally on whether a definition for a given item (“foodstuff”, “wildlife species”, etc.) can be extrapolated to edible insects.

### 3.1. Food Law

As in other regions of the world, the food law of a given African country is determined by both international and national regulations. From the international point of view, Codex Alimentarius, AU and the different African trade blocks are the most important ones. As the Codex is well-known, it will be omitted here.
AU. The *African convention on the conservation of nature and natural resources* is oriented towards managing the entire African environment, regardless of being farmland or not. Member states are called to “establish, strengthen, and implement specific national standards” (p. 10) for processing respectively production methods and product quality (Article XIII).CEMAC. Food security is one of the foci of CEMAC (http://www.cemac.int/node/140). Two institutions are thought to be handling food regulation issues, both localized in Ndjamena, Chad. Access to the homepage of CEBEVIRHA (Commission Économique du Bétail, de la Viande et des Ressources Halieutiques, Economic Commission for Livestock, Meat, and Fishery Resources), was not granted by the authors’ internet due to safety concerns. Pôle Régional de Recherche Appliquée au Développement des Systèmes Agricoles d’Afrique Centrale (Regional Growth Market of Applied Research for the Development of Agricultural Systems of Central Africa, PRASAC) is the other authority, conceived as a network for agricultural research regarding the savannah. No documents could be downloaded.CEPGL. No food-related documents could be encountered on the union’s webpage (http://www.cepgl.org/).COMESA. Chapter 18 of the COMESA treaty refers to the “co-operation in agriculture and rural development” (COMESA, 2009, p. 69), aiming towards “common agricultural policy”, “regional food sufficiency”, productivity increases, and “replacement of imports on a regional basis” (ibid.). In terms of food safety, member states are requested to harmonize their regulation and standards in order to facilitate trade within COMESA borders and cooperate among each other with regard to research incl. that to increase productivity. An early warning system regarding food hazards is also an issue. No more specific documents could be found.CEN-SAD. The web presence (https://web.archive.org/web/20080709020642/http://www.uneca.org/cen-sad/reportofthe 6th ordinarysessionoftheEC.htm) of this trade block mentions that during the 2001 meeting in Ouagadougou, Burkina Faso, opinions regarding food security (cave, not food safety) were exchanged. No more details were available.EAC. This East African REC issued the *EAC food security action plan (2011–2015)*. ECOWAS In it, they include a FAO definition of ‘food security’, i.e., “Food security exists when all people, at all times, have physical, social and economic access to sufficient, safe and nutritious food to meet their dietary needs and food preferences for an active and healthy life” (p. 4). In this way, food security also includes food safety. Apart from improving production, quality, and distribution of common foodstuffs, using an alternative food supply of agricultural, aquatic, and forestry systems is one of the priority areas of this action plan. This would, in fact, include insects, regardless of whether they are caught from the wild or farmed.ECCAS. Like EAC, ECCAS member states agreed to cooperate in terms of agriculture and food supply. Chapter VII of the corresponding treaty (*Treaty establishing the Economic Community of Central African States*) contains these intentions. As other trade blocks, food supply is supposed to be originated from agriculture and livestock, aquaculture, and forestry management, so insects are potentially included.ECOWAS. Article 22 of the ECOWAS treaty (*Revised treaty*) foresees the establishment of a technical commission for food and agriculture. The corresponding section of cooperation in the food sector is Chapter IV. As in other documents, the focus lies on food security rather than food safety.GAFTA. Of this trade union, only the declaration text (*قرار المجلس الاقتصادىإعلان منطقة التجارة الحرة العربية الكبرى والاجتماعى رقم 1317 د.ع 59 بتاريخ 19/2/1997.*) was available. It focuses on the trading conditions of Arab products claiming that a certificate of origin is mandatory, but that imported GAFTA goods will be treated as local ones throughout the member states. It may be assumed that this also includes foodstuffs, but this product was not mentioned in this document.IOC. Like other transnational organizations, food security is one of the primary goals of IOC. Comparable to IGAD, the provided IOC documentation [12] focuses on specific projects to improve food security. However, no legal documents could be encountered.IGAD. Food security and food safety are primary targets of IGAD, being part in several policy statements and programs published (*Regional Strategy, Volume 1 & 2*) rather than concise legal bases.MRU. Due to technical reasons, the organization’s homepage could not be accessed.SACU. Founded in 1910, SACU is the oldest customs union of the world. It recognizes the predominance of the agricultural sector, which plays an important, but variable role in the member states [13]. However, no specific documents could be encountered.SADC. In its declaration of agriculture-related actions to take (*Dar-es-Salam declaration on agriculture and food security in the SADC region*), only aquaculture and “short cycle stocks” (sic!; poultry, small ruminants, pigs) are mentioned. In fact, many insects would be short-cycle livestock. In view of the time of publication, which was far before the FAO started to promote edible insects, opening this definition to productive insects may be an option to be considered, moreover as some insect products like mopane worms (*Gonimbrasia belina*) are an important product with transregional trading relevance. However, the protocol on wildlife management (*Protocol on wildlife conservation and law enforcement*) does include all animal and plant species and allows a sustainable use of this wildlife, particularly by local communities. However, no specific rules were laid down. The guidelines for the regulation of food safety in SADC states (*Regional guidelines for the regulation of food safety in SADC member states*) contain a comprehensive description on how the national food law should be in the 15 member states. Although it does not mention insects as such, the definition of food (“…any substance […] intentionally incorporated into the food…” with a list of duly prohibited substances) would support the inclusion of insects. Many aspects of this document remind on the EU food law, particularly *REC (EC) 178/2002*, but as that latter, no specific data on the different food types or the evaluation criteria is provided.UEMOA. Unlike other trade organizations, UEMOA published a concise regulation of food security and safety. Considering that according to *Réglement 007/2007/CM/UEMOA*, every animal species can be a foodstuff, insects are potentially included. Chapter II deals with the health monitoring of animals and products made thereof. While Section 1 sets the obligations of the involved actors, Section 2 refers to animal health control and inspection, and Section 3 to animal and animal product movements inside and outside member countries. Chapter III attends food safety with a similar layout (Section 1: obligations, Section 2: control and inspection, and Section 3: product movements). In terms of food safety, Article 84 is particularly interesting since it prohibits the production and the placing on the market of foodstuffs that are hazardous to human and animal health, do not respond to the consumer information requirements, do not fulfill the ethics of the international food trade as established by the Codex Alimentarius, or do not cope with the requirements for novel foodstuffs. The latter refers to foodstuffs unknown to the member states’ population and consumed marginally and also includes GMOs. Furthermore, the regulation refers to the so-called Secrétariat Régional de la Normalisation, de la Certification et de la Promotion de la Qualité (NORMCERQ) that is in charge of harmonizing the corresponding regulations, but no hint of this activity regarding food could be encountered.UMA. The Arab Maghreb Union has a commission for food safety, which, according to its website, met 2007 in Nouakchott, Mauretania. It has ratified several conventions regarding the exchange and quarantine of agricultural products among member countries. However, these documents are not online.

As can be seen, most of these international organizations do not set a concise legal framework for the member states. This is a difference to other trade blocks, e.g., the EU. Instead, policies are proposed. Policy documents are typically placed on consulted ministries’ homepages, dealing with ways to increase food security, food safety, intermember states exchange, and food and production-related research. In some cases, policy statements are presented rather than concrete legal frameworks.

Policies seldom address edible insects directly. Some exceptions will be presented on a national level.

On a this level, no data on the food law in Chad and Lesotho could be encountered. Technical and safety reasons made it impossible to access the corresponding web presences in Botswana, Cameroon, Sierra Leone, Sudan, Tanzania, and Togo. FAOLEX was used whenever possible.

No country has explicitly included edible insects in their current food legislation so far. However, direct contact to authorities and consulting the internet provided the following results:According to the consulted authorities, edible insects are tolerated in Algeria, Benin, Burkina Faso, Burundi, Cameroon, Central African Republic, Madagascar, Morocco, Namibia, South Africa, Togo, and Tunisia. All these countries lack a specific regulation for edible insects.In Congo-Kinshasa, Guinea, and Niger, entomophagy is also tolerated, but during the interviews, an interest in establishing a corresponding legal framework was expressed.Benin. The central laboratory for food safety (http://www.lcssa-benin.org/index.php/resultats-analyses/) offers a series of chemical, microbiological and contaminant-related analyses for all foodstuffs. By extension, this would theoretically also include edible insects.Cape Verde. In *Lei n^o^ 30/VIII/2013*, “animal” is defined as either a mammal, bird, or bee (Art. 3). From that, “products of animal origin meant for human consumption” derive. In this way, other insects are excluded, making edible insects illegal on the archipelago. The only exception would be bee brood.Comoros. Food safety is addressed in the public health code (*Code de la santé publique et de l’action sociale pour le bien être de la population*), providing a definition for food hygiene, but not for food.Gabon. Most food legislation is about fishery products.Gambia. The FSQA homepage has a section for regulations, but instead of providing the original texts, bullet points to specific issues are presented of which none applies to edible insects. Within the subsection “standards”, a Food Safety and Quality Act from 2011 is mentioned, but could not be found neither on the FSQA nor the FAOLEX pages.Ghana. The FDA homepage contains a large set of regulations and codes for many sectors of food policy, and there is a list of different foodstuffs and the parameters they should be analyzed for (https://fdaghana.gov.gh/index.php/certificate-of-analysis/), yet omitting the threshold values. None refers to edible insects.Guinea. In fact, most regulation concerns fishery products.Madagascar. On 901 pages, MAEP has merged all relevant regulations pertaining agriculture, animal breeding, and fishery into one document. There is no specific Malagasy legislation for edible insects, but they are accepted, and the Codex Alimentarius is used when determining the hygiene and edibility of insects. However, Codex specifications apply to foodstuffs in general or to specific, non-insect foodstuffs.Malawi. A national policy that promotes insect consumption is said to exist, but could not be retrieved on the internet.Mozambique. While there is an extensive regulatory framework for fishery products, a general food act could not be encountered.Namibia. A new policy for food safety [14] does mention edible insects specifically, postulating that the Ministry of Agriculture, Water, and Forestry should be responsible to elaborate standards for a series of foodstuffs including insects. This policy was submitted to the cabinet by the Minister of Agriculture, Water, and Forestry, but has not been amended yet.

These results show that edible insects are tolerated in many countries, and although there is no regulation for them at the moment, some governments are interested in developing them.

When working with legal texts, it is important to check the definitions provided there. In the case of the definitions for ‘foodstuff’, all countries would permit considering insects as such (Table 4), at least in the case of the countries for which such a definition could be encountered. In most cases in which countries apparently lack a foodstuff definition, however, they do have food legislation, but either they work without the preliminary definition or address specific foodstuffs, e.g., meat, fishery products, or eggs. For UEMOA member states, the food definition provided there would have an official character, even in the countries apparently without their own definition, i.e., Burkina Faso and Senegal. Another probable reason for information gaps may the fact that some regulations have simply not been published on the internet. To give an example, a central regulation for food in Togo is the *Arrêté interministériel n^o^ 06/08/MAEP/MEF*. It is mentioned in secondary literature (e.g., [15]) but could not be traced back in the net.

Although these definitions basically include edible insects, a further insertion into food law is not possible. This is due to the fact that food inspection framework usually includes a definition of the food category, e.g., ‘meat’, ‘fishery product’, etc. With insects being foodstuffs of animal origin, the term ‘animal’ is defined, if at all, in terms of livestock and species obtained via fishing, sometimes including game species. Interestingly, some regulations seem to make a difference between “animal”, “bird”, and “fish”, suggesting that “animal” as a food provider is perceived as a mammal. In fact, *The Food and Drug Act* (Art. 1) from Uganda excludes birds and fish from its ‘animal’ definition. In any case, these definitions usually exclude invertebrates. Some, however, mention honeybees, e.g., *Lei n^o^ 30/VIII/2013* of Cape Verde. One of the relatively few countries that actually mentions insects in the animal definition is the *Food Act, 2014*, of the Seychelles (Art. 2). The *Veterinary Law Code* of Somalia (Section 1) does not contain an overall definition of foodstuff, but for ‘animal product’, an ‘animal’ being “any vertebrate or invertebrate animal other than the human being”.

Some nations also have addressed the idea of novel foodstuffs. To provide an example, Niger’s *Décret n^o^ 2011-616/PRN/MEL* defines them as “products or foodstuffs for which human consumption in Niger has so far been unknown or marginal, as well as foods and ingredients made from genetically modified organisms” (Art. 7/2). A definition for novel foodstuffs is also provided by *007/2007/CM/UEMOA*. It may be debated, however, if edible insects actually are a novel food in African countries, particularly if marginal consumption is used as a criterion. However, as can be seen in Table 3, the amount of edible insect species varies from country to country, as will the extent of traditional entomophagy. In this way, the issue of insects as novel food will have to be answered on the nation level, also for those cases in which edible insects are traded from one country to another.

### 3.2. Wildlife Resources Management

Africa is known worldwide for its spectacular wildlife, which has been exploited for centuries as game, providing meat and trophies. Fearing the extinction of these valuable resources, most countries have developed a comprehensive regulatory framework to manage these species. Laws pertaining wildlife were issued in order to regulate the usage of the large African animals, particularly mammals, birds, and reptiles such as crocodiles, be it as a trophy obtained via sports hunting, be it as a foodstuff during regular hunting. Hunting legislation basically refers to the species that can be hunted, the methods permitted, and the fee to be paid for the right to hunt under a combination of the aforementioned parameters. Inclusion of arthropods such as insects and arachnids must be seen as incidental and a consequence of the definition for ‘animal’, particularly if regulations date from the previous century.

Then, and in view of the many other natural resources, it became necessary to regulate the management of, e.g., wood or mineral resources, i.e., activities that compromise the integrity of certain areas. Finally, natural reserves attract tourism, and so, regulation of these also became necessary.

AU has issued the *African convention on the conservation of nature and natural resources*. It addresses the need of balancing environment protection with sustainable use of the natural resources. In Article V, some basic terms are defined, e.g., “natural resource”, “species”, “specimen” (which refers to any non-human life-form, dead or alive), “product” (any part or derivative of a specimen), and a series of conservation areas types are recognized (defined more precisely in Annex 2). Among others, Article XIV addresses the sustainable use of these resources and Article XVII the traditional rights of local ethnics to make use of them. In Article VI on land and soil, the term “sustainable farming and forestry practices” is coined and considered a goal to follow. Managing these specimens is the basic objective in terms of species diversity (Article IX), be it inside conservation areas, be it outside of them. In the latter case, harvestable populations are allowed. Among other objectives, the introduction of non-native species is to be controlled strictly, and existing populations should be eradicated, pests controlled, and diseases eliminated. This goes along with the phyto-sanitary convention which basically seeks to minimize the impact of, among others, insect pests by restricting the importation of potential sources. Member states are encouraged to regulate the extraction of specimens, avoiding a series of extraction methods specified in Annex 3. Although it may be expected that this passage refers to vertebrates, some of these banned means are used for regular insect hunting, e.g., artificial light sources, target illumination devices, nets (with exceptions as agreed upon by the Conference of the Parties), and traps. It calls the member states to regulate the domestic trade with, transport of, and possession of specimens (Article XI), to ensure the sustainable management of natural resources within development plans (Article XIV), to respect traditional rights and knowledge (e.g., of farmers; Article XVII), enable research (Article XVIII), and promote capacity building and technology transfer (Articles XIX and XX). Still, this paper did not contain concise regulations, but should be seen as a general framework from which other (inter)national regulations may start from.

Below the AU, one of the larger organizations is the Commision des Forêtes d’Afrique Centrale (COMIFAC), which includes Burundi, Cameroun, Central African Republic, Chad, Congo-Brazzaville, Congo-Kinshasa, Gabon, Equatorial Guinea, Rwanda, and São Tomé and Príncipe. In their constituting treaty, they expressed their commitment to support and develop a sustainable use of the forest resources, creating income from these resources (Article 1).

Of all these bases, edible insects only play a minor role, being one of the many forest products that may, or may be not, harvested. The main focus is clearly the game species, either for sport hunting or for bushmeat. Besides, the permitted use of insects (trophy, foodstuff, feedstuff, medicine, etc.) is frequently not stated. In this way, the first approach of asserting the legal status of insects within this area was to determine if insects actually were (respectively could be) included in the ‘animal’ definitions these regulations provide (Table 5). In some cases (e.g., Burkina Faso), the regulation did contain a concise definition, suggesting that ‘animal’ was defined using common sense, i.e., a non-human member of the Animalia kingdom. If this is actually the case, then the *Décret n^o^ 96-061/PRES/PM/MEE/ MATS/MEFP/MCIA/ MTT* would in fact be applicable to insects as it refers to “animals”.

In other cases, regulations exclude insects. To give an example, *Loi n^o^ 1/17 du Septembre 2011 portant Commerce de Faune et de Flore Sauvages* from Burundi refers, in terms of animals, to a list of mammal, bird, reptile, and plant species.

Wildlife can be managed or protected. In most countries, wildlife protection is mainly focused on large animals, typically hunting species. Some nations, however, also include insects. The *Décret n^o^ 2004-0065/PR/MHUEAT* of Djibouti puts all national animal species under protection, i.e., also the insects. In Cape Verde, the *Decreto-Regulamentar N° 7/2002* lists several beetle species as protected, i.e., the scarabids *Aphodius* spp., the hydrophilids *Berosus* spp., the buprestid *Chrysobothris dorsata*, the dytiscid *Eretes sticticus*, the cerambycids *Xystrocera* spp., and the tenebrionids *Alphitobius laevigatus, Tenebrio* spp., and *Zophobas* spp., which are all edible. So, local consumers have to take care not to feed on these species.
Cape Verde. No regulation on wildlife management could be found. However, the *Decreto-Regulamentar n^o^ 7/2002* classifies the national fauna into several types, with different levels of protection. It contains a large list of beetles, of which according to [11] some are edible, at least at the genus level. Unless bred in captivity, these species may not be extracted from the wild or managed in any commercial way (Article 9). For the commercial use of endemic species, a permit from the government is mandatory. Introducing exotic species (as farming insects would be) can only be done after authorization from the government (Article 13). As mentioned before, the list also includes the classical tenebrionid genera *Alphitobius*, *Tenebrio*, and *Zophobas*. Although the text refers to *Tenebrio guineensis*, all *Tenebrio* species are considered edible. The mentioned “Zophobas atratus concolor” is basically *Z. atratus*. This one and *Alphitobius laevigatus* are typically used for food. Installing tenebrionid farms on Cape Verde may therefore be a challenge because the introduction of foreign populations, certifying the captivity origin of national populations or catching the initial stock from the wild are all subject to authorization.Congo-Democratic. The situation appears unclear. Unlike other nations, “non-wood forestry products” as defined in the forest act (*Loi n^o^ 11-2002*) only includes plant products. Wildlife management seems to be regulated by the national hunting law (*Loi n^o^ 82-0022*) and derived acts. Its definition of “game” refers to all vertebrates but amphibians and fish (Article 2) in contrast to “fauna”, which includes both vertebrate and invertebrate species. The rest of the law only refers to game hunting issues. However, the *Arrêté interministèriel n^o^ 003/CAB/MIN/ECN-EF/2006* and *n^o^ 099/CAB/MIN/FINANCES/2006*, which fix the financial framework for wildlife usage deals with “fauna” and also includes “cocoons”, “nymphs”, and “dead insects”. They are part of the “sub-product” section of the partially protected animals’ tax table. It is suggested that these sub-products refer to ornamental insects sold as dried specimens rather than collecting edible insects for consumption. Still, the implementing regulation *Arrêté n^o^ 014/CAB/MIN/ENV/2004* includes specifications for capturing (Article 23) and keeping wild animals (33, 34, and 37–40), i.e., animals in terms of “fauna” and this would include insects. These specifications require regular veterinary controls.Djibouti. Not much could be found on wildlife management. *Loi n^o^ 43/AN/83/1re L* has prohibited any wildlife management for a period of ten years, this law dating from 1983. The more recent *Décret n^o^ 2004-0065/PR/MHUEAT* on biodiversity protection follows this idea, prohibiting any kind of wildlife management.Gabon. There is an interesting, publically available legislation on wildlife management. The forest code (*Loi n^o^ 016/01*) regulates the management of forest resources of which insects are part of. Subject to several permits, the fauna can be used economically and also within traditional frames for basic necessity requirements. *Décret n^o^ 18/PR/MEFEPEPN* regulates the establishment of wild animal breeding units to produce, among others, foodstuffs (Article 2), while *Décret n^o^ 692/PR/MEFEPEPN* focuses on the customary usage of forest products. A position paper [16] promotes a sustainable usage of these resources for the coming years.Guinea. The *Loi U97/038/An* has been clearly made for vertebrates and, in particular, game species. However, insects also fit into the “wildlife fauna” definition, and Chapter 9 regulates the use of unprotected species (a concrete list of the protection status is, however, not provided, and including insects in this section is an assumption by the authors), claiming that handling up to five specimens of the given species at a time is allowed, but contacting the local forest inspector is mandatory if this number is exceeded. This would be the case in harvesting edible insects.

Finally, a modern way of handling insects is by farming them. A survey conducted via FAOLEX in search of applicable data did not yield any results.

So, edible insects can be handled in many African countries in a traditional way, but typically after an official permit was provided.

### 3.3. Edible Insects in Pest Legislation

Some pest insects are edible, e.g., mealworms or locusts which, when swarming, are responsible not only for marked economic losses but may also trigger famines. Thus, there are other joint ventures in which several African nations cooperate. Two of them are CLCPRO (هيئة مكافحة الجراد الصحراوي في المنطقة الغربية—Commission de Lutte contre le Criquet Pèlerin dans la Région Occidentale, Commission for Fighting the Desert Locust in the Western Region) and CRC (هيئة مكافحة الجراد الصحراوي في المنطقة الوسطى, Commission for Controlling the Desert Locust in the Central Region), hosted by FAO. Both commissions are dedicated to preventing damages to crops caused by swarming desert locusts (*Schistocerca gregaria*). The use of this natural resource as foodstuff has not been contemplated so far, mostly out of the sheer size of locust swarms, and so no regulation in this issue has been established.

Besides, most African nations have more or less detailed pest legislation. There is usually one basic regulation that defines pests in a way that insects are included (Table 6) and provide measures on what to do to ensure pest reduction. In addition, many countries have specific acts dealing with plant diseases and pests in which the corresponding insect species are listed (Table 7, Table 8, Table 9 and Table 10). To give an example, Morocco issued an interesting regulation for the control for the African palm weevil (*Rhynchophorus ferruguineus*). This regulation makes fighting this weevil obligatory, and a Moroccan governorate was classified as a quarantine zone, forbidding all transport of palms out of this area. Palms affected by the weevil must be destroyed (*Arrêté conjoint du ministre de l’agriculture et de la pêche maritime et du ministre de l’intérieur n^o^ 287-09 du 30 janvier 2009 édictant des mesures d’urgence destinées à la lutte contre le Charançon rouge du palmier* (*Rhynchophorus ferrugineus*)). However, for the scope of the present contribution, the fact that these species are not permitted is the decisive information rather than the measures taken to control these pest species. In this way, rearing and placing on the market this particular species would be against the law.

Swarming locusts have been a threat for the agriculture for millennia. Along with the CLCPRO and CRC initiatives, many countries have adopted national regulations. However, only some do actually include concrete measures. In other countries, the fight against locusts is documented legally by the will to build up corresponding organizations and initiatives, i.e., the national anti-locust agency (ANLA) in the Chadian *Loi n^o^ 005 /PR/2007*, IFVM in Madagascar (*Décret n^o^ 2017-064 du 31 du janvier 2017*), the CNLP in Mali (*Loi n^o^ 06-065 du 29 décembre 2006* and subsequent texts), the PLUCP in Niger (*Arrêté n^o^ 13/MDA/DPV*), and the campaigns in Morocco (*Loi n^o^ 57-02*) and Tunisia (*Décret n^o^ 88-1751*). In those countries, locust may be combatted efficiently, but this is not reflected in the law.

While locusts, butterfly caterpillars, and beetle grubs and larvae are relative well-known food insects, some of the species contained in Table 10 are not, particularly the cockroach and fly species. It should be stressed that the species listed are known to be edible, at least in some parts of their ranges, which may not necessarily be Africa. The record of edibility of *Periplaneta americana* comes from Brazil, that of *Anastrepha ludens* from Mexico.

As a summary of the previous tables, Table 11 provides an overview of the current legal situation of edible in the different African nations.

## 4. Discussion

### 4.1. Data Availability

One of the main difficulties in carrying out the present survey was data gathering. Getting first-hand information from the corresponding authorities was challenging. However, as the group of authors comprised African and European scientists, the African colleagues had better chances in getting in contact with these authorities. A clear cut was seen between the responses of countries where French and Arabic are spoken (attended by the African colleagues) and those in which English or Portuguese are the official languages (attended by the European colleagues), with the latter responding very late, if at all (despite several attempts made to get an answer), while the African colleagues were able to provide results within two weeks. Differences between the authorities and citizens asking for their advice is a global phenomenon, but also African researchers make special emphasis on the local conditions [17].

Data availability is also an important issue on the internet; some countries did not provide their current legislation on the corresponding government portals. Instead, data had to be traced via other sites, especially FAOLEX. Still, gaps remained, either due to technical or internet safety reasons, or simply because some governments do not put these sources online.

In this context and beyond the current survey, it was noted that most regulatory texts were written either in French, English, or Portuguese, i.e., the administrative languages from colonial times. The only exceptions were the regulations presented (sometimes exclusively) in Arabic of the North African countries, and the full set of regulations from Ethiopia and Rwanda, which were either bilingual (Amharic and English) or even trilingual (Kinyarwanda, English, and French). While this was rather advantageous for the researchers, this may pose a problem for the local African citizens. In Africa, there are more than 2000 languages spoken, and in many countries, the percentage of the population that understands these official languages completely may be relatively low. This, of course, varies strongly among countries, ranging between 35% (Mali) and 88% (Eswatini) in 2018 (http://data.uis.unesco.org/Index.aspx?queryid=166#). So, there is a risk that a certain portion of a country’s population may not be able to understand the laws of their own country. However, the situation is changing in some countries. The web presence of the government of South Africa (https://www.gov.za/) has been presenting regulations in several of its eleven official languages, usually English plus another one.

### 4.2. Edible Insects in African Food Legislations

With the exception of phane in Botswana, edible insects have not been considered in the African food law, neither on the national nor on international level. This may be surprising in view of the rich entomophagy tradition that exists in Africa. Van Huis [6] stressed the importance of edible insects in the different regions of the continent, which range from using them in those seasons of the year in which other natural foodstuffs are rarer to the simple reason that insects are consumed because of their taste. This is a completely different scenario from, e.g., the EU where insect-consuming tradition is almost inexistent, and insects are treated as truly novel foods that enter the European food market. In this way, the EU novel food regulation opens this market, and future legal acts will incorporate insects into the public health surveillance, adopting as much as possible from existing regulation, but attending, at the same time, the idiosyncrasies of these foodstuffs, e.g., microbiological criteria. This may be flanked by national European guidelines [7]. However, food law in Africa by itself is the key to this finding.

It should be stressed that, on one hand, African food legislations are merely starting to adapt to the current standards experienced in other parts of the world, and that, on the other hand, the progress in that varies markedly among countries. Some laws cited in the present contribution date down to the 1900s and are still in power. Yet, these are exceptions, and many laws date from the 1990s to the 2010s. In some countries, no food law could be encountered, and if available, many laws seem to lack a concrete depth. In some way, many of these texts express expectations about the issues the address, leaving a very large range for interpretations, making law less transparent. However, this phenomenon is far from being exclusively African. Many EU laws also contain these kinds of expectations, e.g., *REG (EC) 178/2002*. However, subsequent acts provide the details of the bases set in these general laws, e.g., *REG (EC) 2073/2005* on the microbiological criteria for a series of foodstuffs. Still, the current EU novel food regulation (*REG (EC) 2015/2283*) also sets the bases for novel food certifications, but does not contain specific indication on production, processing, or quality control. A corresponding amendment of *REG (EC) 2073/2005* for the EU is dearly awaited.

So, African food laws are sometimes outdated and lack depth, which would increase their applicability. However, food safety in Africa is a basic problem, regardless of how efficient the legislation is. Kussaga et al. (2013) [18] present a large list of microbiological and chemical risks detected in African foodstuffs, ranging from pesticides, mycotoxins, heavy metals, over many bacteria to a series of vermiform parasites. Apart from that, Africa is experiencing a serious problem with antibiotics residues in its foodstuffs. An extensive review concluded that in some African countries, the prevalence of veterinary drugs contained in foodstuffs amounted to up to 94% in certain countries. This may affect both the food-processing industry (e.g., fermented dairy products) and the consumer alike [19]. This shows that food safety in Africa is a very immanent problem.

This condition becomes even more complicated. On one hand, traditional food habits have changed in Africa, creating products not clearly addressed in the current legislation. As an example, for Western Africa, not only the total amount of food required has increased, but consumers’ preferences have shifted to convenience products, consuming more perishable products, and a growing awareness of food quality issues [20]. Improving policy coordination and policy implementation are, in fact, two of the key recommendations the aforementioned authors give to modernize the Western African food sector.

On the other hand, the “modernization” of traditional foods does not always improve their quality. As traditional food production has experienced labor ease by means of mechanization of e.g., cutting and grating processes, the production increase has not lead to a growing awareness of food safety, good manufacturing practices, and poor hygiene. Selling food that requires a certain degree of air circulations in tight-sealing plastic bags may also affect the quality by favoring the growth of microaerophilic or even anaerobial flora [17]. Since insects are traditional foodstuffs, these risks may also apply to them.

The degree of entomophagy also varies across the continent. Table 3 shows the different amounts of REIS per country, and while insect consumption may be rather unimportant in e.g., Djibouti or São Tomé and Príncipe, in Botswana, Benin, or Madagascar it plays a larger role. Besides and strictly speaking, introducing an edible species from one African nation to another where the consumption is unknown would make this species a novel food as coined in UEMOA, Niger, or EU regulation.

Another important issue is to tell food safety from food security. Strictly speaking, food security refers to the supply with (any kind of) food (in any quality), while food safety, in contrast, means to provide a foodstuff in a quality that does not compromise the consumer’s health, i.e., free of pathogens, toxic substances, foreign bodies, radiation, etc. In many countries, food security is considered more important, particularly in those in that climate and other factors may lead to food shortages. However, AU documents include food safety in food security. Given the importance of food insects in times of the year when food is scarce (the rainy season, time before harvest, etc. [21]), the lacking of the consideration of insects still remains interesting. A possible reason is the fact that a certain portion of the current food legislation originates from colonial times, and the omission of edible insects in the first place may be part of the entomophobia of the colonial rulers. In fact, many people in Europe still consider eating insects either disgusting or a staple for the poor and primitive that cannot afford any “true” food” [22], although these concepts do not, by any chance, reflect the reality. In fact, this seems to be one of the major reasons why entomophagy is still not more popular in Europe, despite corresponding awareness campaigns.

All in all and in comparison to other areas of the world, food safety regulation in many African countries appears scarce or even not existing. One possible explanation may be the fact that governments were forced to focus on food security rather than food safety in response to natural catastrophes and warfare. However, as international African documents exposed here showed, the idea that food security mandatorily has to include food safety is becoming increasingly attended. Besides, the AU has underlined that poor food safety is not only a problem for the population (with 91 million cases of food-borne diseases in Africa per year and 137,000 casualties related to them) but also a problem decreasing the competitiveness of African agriculture inside and outside the continent [23]. The goals formulated in the Malabo Declaration to end hunger in Africa by 2025 will not be reached if food safety is left unattended. In response to that, the first FAO/WHO/AU International Food Safety Conference was held in Addis Ababa in February of 2019, and edible insects were addressed in one session [24], underlining the need to establish concrete policies also with regard to farming insects since the absence of specific regulations may trigger the establishment of insect farms that use—in terms of food safety—unsafe substrates, e.g., slaughterhouse wastes. The ideas raised in Addis Ababa were presented again at a subsequent meeting in Geneva in April of 2019, where the need of harmonizing food safety regulation was stressed [25].

Taking up the misleading conceptions that appear in Western societies on entomophagy, the reality in Africa is, at least locally and seasonally, a huge variety of vivid entomophagous traditions, far from being an “emergency foodstuff” (Figure 1 and Figure 2). Most government bodies interviewed directly consider this tradition, tolerating the consumption and the trade with these animals, even though no concise framework exists. Table 4 showed that most African countries work with foodstuff definitions that permit the inclusion of edible insects, although they may not have been considered at the moment of publishing the corresponding regulations. This is a positive fact, and future framework can be based on these definitions. The will of some of the interview partners to fill this legal gap is also a promising perspective.

This was also confirmed by the minutes of a meeting of several governmental and non-governmental organisations (GO and NGO) from in and outside Africa, held in Kenya [26]. One of the meeting’s goals was to create awareness of the importance of the insect sector and the need to establish a corresponding legislation, which ideally could be harmonized among countries to ensure a comparable quality. These regulations should include both food safety standards and best practice codes. The future will show what will become of this initiative.

### 4.3. Food Insects Obtained from the Wild

Following the tradition, insects have been caught from the wild in order to be processed and consumed [5]. African wildlife management legislations referring to hunting and/or natural resource management (as that of forests) usually base themselves on “wildlife” definitions that do include insects. In this context, and strictly speaking of law, this can result as a disadvantage, particularly in the moment of hunting respectively gathering them. As in the food law, the application of wildlife management laws to insects may result by mere extrapolation as the regulations were written while having traditional livestock or game species in mind. In those, using some trapping and hunting techniques is clearly banned. However, they do apply to what the definitions understand as “wildlife”, and so, light traps, which are typically used to attract insects in the night, may be in fact illegal. The same is true for the permits many regulations request. It may be expected that, except phane in Botswana, gathering edible insects from the wild will be done without asking for these permits, perhaps because it is a traditional way of eating and earning money, perhaps because insects do not have the perceived relevance as cattle or high-priced game species.

Still, forest and wildlife management offer an interesting way to use the insect resource in a sustainable way. These species may be managed in their corresponding biotopes. This, however, requires the awareness that only sustainable use of these insects will ensure a longer-lasting income, rather than exterminating a species by over-harvesting. For this, reliable data on the degree to which a population may be extracted without endangering its survival are necessary, and this data used to be inexistent for insects. Only local gatherers may have this knowledge out of their experience.

In any way, Table 5 shows a large degree of variation among countries. Some allow wildlife management, some do not, and some regulations only refer to specific animal or product groups. Due to the complexity of this issue, it is highly recommended to study the actual situation in a given country intensively before any major uses of insects are made. Tchibozo et al. [27] summarize the necessary requirements to use wild populations thoroughly:Good knowledge of the edible species of a given region;Thorough species inventory of the area in question;Identification of the edible species;Good knowledge of the species’ biology including their host plants;Detection of breeding sites;Detection of places where they are/can be modified.

With knowledge, the particular breeding behavior and the feasibility for production can be assessed. This model is laid out for satisfying a local market and is the base of obtaining a solid colony for breeding in captivity, with the option of selling the offspring also abroad [27].

Besides, food safety concerns remain, since forest-derived foodstuffs are even less subjected to quality controls than livestock-derived ones are. Although tradition is prone to cope with traditional food safety problems like spoilage or pathogens, more modern risks as contaminants like insecticides are likely to pass undetected. Besides, modern storage methods like plastic bags also may impair the quality of insect products sold on the market.

A special situation is those species of insects that are edible and a pest at the same time. The idea of consuming edible pest insects is more than tempting, particularly since many of these species have a better nutritional profile that than the crops they feed on. To give an example, teff (*Eragrostis tef*) is a typical East African crop, and contains 12 g of protein/100 g dry matter (value calculated from Baye [28], while orthopterans range between 41 and 91 g/100 g dry matter [6]. Using pest insects may be beneficial from the nutritional point of view opening up a new food resource; it reduces the damage to the crops and is, by itself, part of the environmental management of a given area. Legally, this may be difficult as many countries have established corresponding legislations to combat them. This applies mainly to locusts, but also to other species, as can be seen in Table 7, Table 8, Table 9 and Table 10. From the food safety point of view however, the proper use of this food resource will depend fundamentally on the toxicological status of these animals (pesticides) and the possibility to process these animals right after harvesting into a storable product with a stable quality. In fact, orthopterans killed by insecticides are sometimes gathered and sold on West African markets [5].

Finally, some countries protect some native insect species from usage, e.g., Cape Verde and Djibouti, including some that are edible. This is a regular proceeding to protect the national fauna, particularly if the territory is relatively small and, in the case of Cape Verde, a set of island relatively far off the continental coast.

Thus, managing edible insects in the wild is an ancient tradition. However, a sound legal system to ensure sustainability will be necessary to make a larger use of edible insects.

### 4.4. Insect Farming

When wildlife management may be difficult because of legal uncertainties and food safety concerns, farming them may be an option [4]. Southeast Asian countries like Thailand, Cambodia, Laos, and Myanmar have started a larger attempt to rear insects and produce them as “mini-livestock” [29]. Two so-called Thai Agricultural standards that deal with crickets (*TAS 8202-2017*) and silkworms (*TAS 8201/2012*) are examples on how insect farming is regulated by the government. This option is basically also open to African countries.

From the legal point of view, livestock regulations in Africa are even scarcer than food-related ones and refer, if available at all, exclusively to classical and local domestic species.

Some countries allow wildlife respectively game farms, and if definitions fit, they could be nominally also applied to insects (Table 5). However, the choice of indigenous farmed species must be done carefully in order not to infringe any species protection, wildlife management, or pest control regulation (see above). Bringing foreign productive species (certain cricket and mealworm species or the black soldier fly) into the country may also be hazardous and may infringe on other nature protection or trading laws.

Raheem et al. [4] mention first insect farms for the black soldier fly (*Hermetia illucens*) in South Africa (which in fact is the largest in the world) and Nigeria, but for feed production. Edible crickets are reared in Kenya and Uganda. This implies that at least in those two countries, the setup of these farms was legal.

## 5. Conclusions

In this way, asking for food quality standards for edible insects may be challenging if some countries have not addressed food quality standards for meat, fish, or dairy products so far. However, the world progresses, and Africa is a part of the global community just like any other region of the world is, and if foodstuffs’ quality will be addressed in the future, those for edible insects should be included. The regulatory framework should address the entire production chain, starting from two different points of primary production, i.e., wildlife resource and farmed insects.

Regarding gathering from the wild, many African countries already have a solid legal base that could be applied to insects. However, population control is vital in order to protect existing natural populations. For food safety, the quality of the gathered animals is critical, particularly in terms of residues and contaminants, and harvested animals should be tested on a regular base, e.g., using Codex Alimentarius recommendations.

The possibility of farming insects in Africa must be assessed thoroughly, taking care not to interfere with land ownership, limited possibilities of using a given area, and, if applicable, farming regulations for livestock and/or wildlife. Animal species’ protection, animal imports, and pest legislation must also enter this evaluation. The farmer must be able to control the life cycle of the farmed species. Special care must be taken to avoid the escape of the farmed insects respectively the entry of insect predators into the farm. Once established, a farm ensures food quality by providing a controlled environment to the farmed animals, from the materials used for construction and bedding to the feedstuffs. This is where residues and contaminants can also enter the production chain of a farm. Farms should also be evaluated on a regular base, e.g., to obtain a certificate that it works according to the national requirements. During these evaluations, the inspectors should pay special attention to animal welfare, meaning that the five freedoms also apply to farming insects. Whenever possible, farming must be done according to national law.

Regardless the origin of the harvested animals, killing must be done according to the animal welfare state-of-the-art procedures, which currently is freezing them or mushing them quickly in a blender. As soon as possible, insects must be heated thoroughly (e.g., 100 °C for 10 min) to reduce bacterial counts, and rinsed. Storage of merely cooked insects must be in the refrigerator or an ice box, or, better, frozen. Insects preserved by other methods, e.g., drying, smoking, or fermenting can be stored traditionally. However, microbial counts should be assessed for these products in order to provide best-before dates for the consumer. Besides, general hygiene rules also apply to insects just as any other foodstuff.

In terms of public health surveillance, insects sold publicly, regardless if gathered or farmed, should be included in regular monitoring programs. There is no need to treat insect-based products any differently as more established foodstuffs are. Appropriate labeling follows national guidelines and should, by any chance, show the name of the insect (ideally including the scientific name), a best-before date and the information about allergens; all insects contain tropomyosin, which may lead to cross reactions of patients that are allergic to dust (mites) and crustaceans. Laboratory analyses should focus, on one hand, on the gross chemical composition (this may change from batch to batch if feeding regime was changed between batches), antibiotics (using substance specific tests rather than generic inhibitory ones as many insects contain innate inhibitors, which could lead to false-positive results), residues, and contaminants. The other focus is microbiology, with salmonellae, listeriae, total bacterial counts, coagulase-positive staphylococci, *Escherichia coli* and other Enterobacteriaceae, presumptive *Bacillus cereus*, yeasts, and fungi as the most important parameters, plus the ones established in national food law.

## Figures and Tables

**Figure 1 foods-09-00502-f001:**
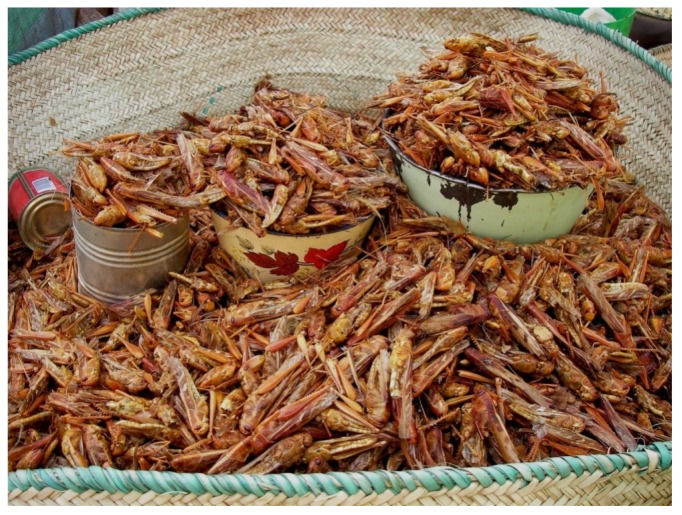
Fried locusts sold on a Western African market (image by S. Tchibozo).

**Figure 2 foods-09-00502-f002:**
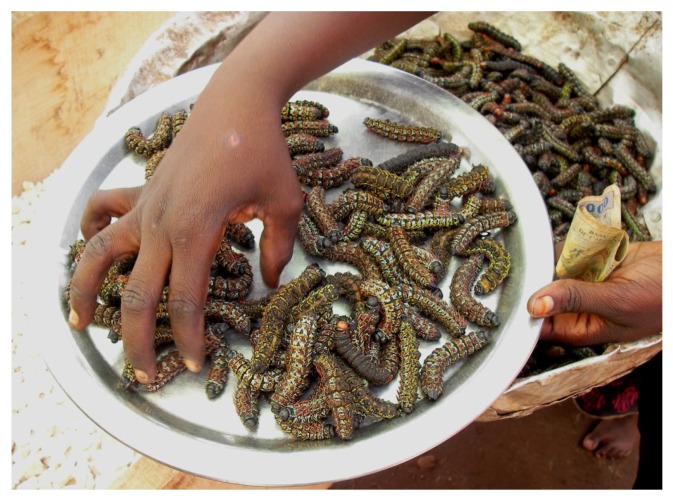
Living caterpillars sold on a Central African Market (image by S. Tchibozo).

**Table 1 foods-09-00502-t001:** Selection of microbiological findings in three African edible insect species (African mole cricket (*Gryllotalpa africana*), cabbage tree emperor moth (*Bunaea alcinoe*), and African palm weevil (*Rhynchophorus phoenicis*)); based on [1]. Blank spaces either mean that the sample was negative for that pathogen or was not tested for it.

Species	Product	*Acinetobacter* spp.	*Bacillus* spp.	*Corynebacterium* spp.	*Enterobacter* spp.	*Enterobacter Faecalis*	*Escherichia Coli*	*Klebsiella* spp.	*Micrococcus* spp.	*Proteus* spp.	*Pseudomonas* spp.	*Serratia* spp.	*Staphylococcus* spp.	*Aspergillus* spp.	*Mucor* spp.	*Rhizopus* spp.	*Saccharomyces* spp.
*Gryllotalpa-africana*	Adults and nymphs, raw		+						+	+			+				
*Bunaea alcinoe*	Larva, raw	+	+						+				+				
*Rhynchopho-rus phoenicis*	Skin		+					+			+	+	+				+
	Gut		+		+							+	+				
	Larva, fresh		+		+							+	+				
	Larva, fried		+										+				
	Larva, roasted		+			+	+				+		+				

**Table 2 foods-09-00502-t002:** Affiliations of African nations to the different trading blocks. See text for abbreviations. Blocks not addressing food issues were excluded.

Country	CEMAC	CEN-SAD	CEPGL	COMESA	EAC	ECCAS	ECOWAS	GAFTA	IGAD	IOC	MRU	SACU	SADC	UEMOA	UMA	WAMZ
Algeria								+							+	
Angola						+							+			
Benin		+					+							+		
Botswana												+	+			
Burkina Faso		+					+							+		
Burundi			+	+	+	+										
Cameroon	+															
Cape Verde		+				+	+									
Central African Republic	+	+				+										
Chad	+	+				+										
Comoros		+		+						+			+			
Congo, Democratic Republic (Kinshasa)			+	+		+							+			
Congo, Republic (Brazzaville)	+					+										
Djibouti		+		+					+							
Egypt		+		+				+							+	
Equatorial Guinea	+					+										
Eritrea		+		+					+							
Eswatini				+								+	+			
Ethiopia				+					+							
Gabon	+					+										
Gambia		+					+									+
Ghana		+					+									+
Guinea		+					+				+					+
Guinea-Bissau		+					+							+		
Ivory Coast		+					+				+			+		
Kenya		+		+	+				+							
Lesotho												+	+			
Liberia		+					+				+					+
Libya		+		+				+							+	
Madagascar				+						+			+			
Malawi		+		+									+			
Mali							+							+		
Mauritania		+														
Mauritius				+						+			+			
Morocco		+						+							+	
Mozambique													+			
Namibia												+	+			
Niger		+					+							+		
Nigeria		+					+									+
Rwanda			+	+	+	+										
São Tomé and Príncipe		+				+										
Senegal		+					+							+		
Seychelles				+						+			+			
Sierra Leone		+					+				+					+
Somalia		+		+					+							
South Africa												+	+			
South Sudan					+				+							
Sudan		+		+				+	+							
Tanzania					+								+			
Togo		+					+							+		
Tunisia		+		+				+							+	
Uganda				+	+				+							
Zambia				+									+			
Zimbabwe				+									+			

**Table 3 foods-09-00502-t003:** List of African countries and authorities that provided information on the legal status of edible insects including the amount of edible species per countries according to Jongema [11]; besides these country-specific data, 4 species were recorded for Northern, 91 for central, 2 for Western, 9 for Eastern, and 20 for Southern Africa.

Country	Institution	REIS *
Algeria	وزارة الفلاحة و التنمية الريفية و الصيد البحري (Ministry of Agriculture, Rural Development and Fisheries)	0
Angola	MINAGRIF—Ministério da Agricultura e Florestas (Ministry of Agriculture and Forests)	16
Benin	ABSSA—Agence Béninoise de Sécurité Sanitaire des Aliments (Beninese Agency for the Sanitary Security of Foodstuffs)	24
Botswana	Ministry of Health	22
Burkina Faso	Ministère de l’Agriculture et des Aménagements Hydrauliques (Ministry of Agriculture and Water Engineering)	7
Burundi	Ministère de l’Agriculture et de l’Élevage (Ministry of Agriculture and Livestock Breeding)	1
Cameroon	MINADER—Ministère de l’Agriculture et du Développement Rurale (Ministry of Agriculture and Rural Development)	59
Cape Verde	Ministério da Agricultura e Ambiente (Ministry of Agriculture and Environment)	0
Central African Republic	Ambassade de la République Centrafricaine à Paris (Embassy of the Central AfricanRepublic in Paris)	54
Chad	Ministère de la Production, de l’Irrigation et des Equipements Agricoles	1
Comoros	Ministère de l’Agriculture, de la Pêche, de l’Environnement, de l’Aménagement du Territoire et de l’Urbanisme (Ministry of Agriculture, Fishery, Environment, Land Use Planning, and Urbanization)	0
Congo, Democratic Republic (Kinshasa)	Ministère de l’Agriculture de l’Élevage et de la Pêche de la République Démocratique du Congo (Ministry of Agriculture, Livestock Breeding, and Fishery of the Democratic Republic of the Congo)	107
Congo, Republic (Brazzaville)	Ministère de l’Agriculture de l’Élevage et de la Pêche (Ministry of Agriculture, Livestock Breeding, and Fishery)	59
Djibouti	MAEM—Ministère de l’Agriculture, de l’Elevage et de la Mer, Chargé des Ressources Hydrauliques (Ministry of Agriculture, Livestock, & Fisheries, in charge of water resources)	0
Egypt	وزارة الزراعة وإستصلاح الأراضي (Ministry of Agriculture and Reclamation of Land)	1
Equatorial Guinea	Ministerio de Agricultura y Bosques (Ministry of Agriculture and Forests)	1
Eritrea	Botschaft des Staates Eritrea in der Bundesrepublik Deutschland (Embassy of the State of Eritrea in the Federal Republic of Germany)	0
Eswatini	Ministry of Agriculture	20
Ethiopia	ጤና ሚኒስቴር - ኢትዮጵያ (Ministry of Health)	1
Gabon	Ministère de l’Agriculture, de l’Elevage, chargé de la mise en œuvre du programme Graine (Ministry of Agriculture, Livestock Breeding, in charge of carrying out the Graine program)	11
Gambia	FSQA—Food Safety and Quality Authority	1
Ghana	FDA—Food Safety Division	3
Guinea	Ministère de l’Agriculture (Ministry of Agriculture)	9
Guinea-Bissau	Ministério da Agricultura e Desenvolvimento Rural (Ministry of Agriculture and Rural Development)	2
Ivory Coast	Ministère de l’Agriculture et du Dévelopment Rural (Ministry of Agriculture and Rural Development)	5
Kenya	Ministry of Agriculture, Livestock and Fisheries	10
Lesotho	Ministry of Agriculture and Food Security	20
Liberia	MOA—Ministry of Agriculture	2
Libya	وزارة الزراعة والثروة الحيوانية والبحرية (Ministry of Agriculture, Fisheries, Livestock, and Irrigation)	1
Madagascar	MAEP—Ministère de l’Agriculture, de l’Elevage et de la Pêche (Ministry of Agriculture, Animal Breeding and Fishery)	34
Malawi	Ministry of Agriculture, Irrigation, and Water Management	20
Mali	MEADD—Ministère de l’Environnement, de l’Assainissement et du Développement Durable (Ministry of Environment, Water Treatment, and Sustainable Development)	8
Mauritania	وزارة التنمية الريفية/Ministère de Développement Rural (Ministry of Rural Development)	0
Mauritius	Ministry of Agro Industry and Food Security	2
Morocco	وزارة الفيلاحة والصيد الباحري والتينمية القريية والميجاه والغابات/Ministère d’Agriculture, de la Pêche Maritime, du Développement Rural, et des Eaux et Forêts (Ministry of Agriculture, Fishery, Rural Development, Water and Forests)	2
Mozambique	SETSAN—Secretariado Técnico de Segurança Alimentar e Nutricional (Technical Secretariate of Food and Nutrition Safety)	3
Namibia	Ministry of Agriculture, Water, and Forestry	27
Niger	Ministère de l’Èlevage (Ministry of Animal Breeding)	19
Nigeria	NAFDAC—National Agency for Food and Drug Administration and Control	19
Rwanda	MINAGRI—Ministère de l’Agriculture (Ministry of Agriculture)	0
São Tomé and Príncipe	MADR—Ministério da Agricultura e Desenvolvimento Rural (Ministry of Agriculture and Rural Development)	4
Senegal	Ministère de l’Agriculture et de l‘Équipement Rural (Ministry of Agriculture and Rural Supply)	10
Seychelles	Ministry of Fisheries and Agriculture	0
Sierra Leone	Ministry of Agriculture, Forestry and Food Security	7
Somalia	Wasaaradda Beerahaiyo Waraabka (Ministry of Agriculture and Irrigation)	0
South Africa	FACS—Food Advisory Consumer Service	58
South Sudan	Ministry of Agriculture and Forestry	1 **
Sudan	وزارة الثروة الحيوانية و السمكية و المراعي (Ministry of Livestock, Fisheries, and Rangeland)	7 **
Tanzania	Ministry of Livestock and Fisheries	22
Togo	Ministère de l’Agriculture, de la Production Animale et de l’Halieutique (Ministry of Agriculture, Animal Breeding, and Fishery)	14
Tunisia	وزارة الفلاحة (Ministry of Agriculture)	0
Uganda	MAAIF—Ministry of Agriculture, Animal Industry, and Fisheries	8
Zambia	Ministry of Agriculture	75
Zimbabwe	Ministry of Lands, Agriculture, Water, Climate and Rural Resettlement	47

* REIS = recorded edible insect species, ** due to its independence in 2011, some species recorded earlier for Sudan may also occur in South Sudan.

**Table 4 foods-09-00502-t004:** Definitions of “foodstuff” in African food regulation frameworks; for Botswana, Burundi, Cameroon, Chad, Comoros, Congo-Brazzaville, Congo-Kinshasa, Djibouti, Equatorial Guinea, Eritrea, Gabon, Gambia, Guinea, Lesotho, Malawi, Mauritania, Libya, São Tomé and Príncipe, South Sudan, and Togo, no definitions could be encountered.

Country	Definition (Regulation)
Algeria	“…any treated, partially processed or raw substances intended for human consumption and including beverages, chewing gum and any substances used in the manufacture, preparation and processing of foods, excluding those used only in the form of drugs or cosmetics.” (*Décret exécutif n^o^ 91-53*, Art. 2)
Angola	“…any substance or mixture of substances, in the solid, liquid, pasty or any other suitable form, intended to provide the human body with the normal elements essential for its formation, maintenance, and development.” (*Decreto Presidencial n^o^ 179/18*, Art. 4 c)
Benin	“…any treated, partially processed or raw material intended for human consumption and includes beverages, chewing gum and all substances used in the manufacture, preparation and processing of foods, excluding those used only in the form of drugs or cosmetics.” (*Loi n^o^ 84-009 du 15 mars 1984*, Art. 2)
Botswana	“…means any animal product, fish, fruit, vegetable, condiment, beverage and any other substance whatever, in any form, state or stage of preparation which is intended or ordinarily used for human consumption, and includes any article produced, manufactured, sold or presented for use as food or drink for human consumption, including chewing gum, and any ingredient of such food, drink or chewing gum” (*Food Control Act*, 2)
Burkina Faso	“…any treated, partially treated or unwrought substance intended for human consumption, and includes beverages, chewing gum and all substances used in the manufacture, preparation and processing of foods, excluding substances used only in form of drugs, cosmetics or tobacco” (*Règlement n^o^ 007/2007/CM/UEMOA*, 1)
Cape Verde	“…any substance or product, processed or partially processed or unprocessed, intended to be ingested by, or reasonably likely to be, human beings” (*Decreito-Legislativo n^o^ 3/2009*, Art. 3.1)
Central African Republic	“…any raw or partially processed substance intended for human consumption” (*Loi n^o^ 03.04*, Art. 24)
Egypt	“…any product or substance intended for human consumption, whether primary, raw, semi-processed, wholly/partially processed or not processed, including beverages and bottled water or food additives and any substance including water and gum, except for fodder and plants and crops before harvest, live animals and birds prior to their transport to slaughterhouses, sea creatures and farm-raised fish prior to fishing, pharmaceutical products and cosmetics” (*Law 1/2007*, (1)/6)
Eswatini	“…means food of animal origin including meat, milk, fish, honey and their products. (*Veterinary Public Health Act*, Art. 2)
Ethiopia	“…means any raw, semi-processed or processed substance for commercial purpose or to be served for the public in any way intended for human consumption that includes water and other drinks, chewing gum, supplementary food and any substance which has been used in the manufacture, preparation or treatment of food, but does not include tobacco and substances used only as medicines” (*Proclamation № 661/2009*, Art. 2, 1/)
Ghana	“…includes water, a food product, a live animal or a live plant, and (a) a substance or a thing of a kind used, capable of being used or represented as being for use, for human or animal consumption whether it is live, raw, prepared or partly prepared, (b) a substance or a thing of a kind used, capable of being used or represented as being for use, as an ingredient or additive in a substance or a thing referred to in paragraph (a), (c) a substance used in preparing a substance or a thing referred to in paragraph (a), (d) chewing gum or an ingredient or additive in chewing gum or a substance used in preparing chewing gum, and (e) a substance or a thing declared by the Minister to be a food under Section 146 (3)” * (*Public Health Act, 2012*, Art. 149)
Guinea-Bissau	“…any substance, whether or not treated, intended for human consumption, by swallowing beverages and products of the type of chewing gum, as well as the ingredients used in its manufacture, preparation and treatment” (*Decreto n^o^ 62-E/92*, Art. 2)
Ivory Coast	“…any food, product, or drink intended for human consumption.” (*Décret n^o^ 92-487 du 26 aôut 1992*, Art. 2)
Kenya	“…includes any article manufactured, sold or represented for use as food or drink for human consumption, chewing gum, and any ingredient of such food, drink or chewing gum” (*Food, Drugs and Chemical Substances Act*, 2.)
Liberia	“…articles including liquids used as nutriment for human consumption or use, alcoholic and nonalcoholic beverages, chewing gum, ice and articles used for components of any such article. *(Public Health Law—Title 33—Liberian Code of Laws Revised*, § 26.1)
Madagascar	“…any substance or product, processed, partially processed or unprocessed, intended to be ingested or reasonably likely to be ingested by humans. This term also includes beverages, chewing gums and any substance, including water, intentionally incorporated into food during their manufacture, preparation or processing.” (*Loi n^o^ 2017-048*, Art. 2)
Mali	“…any totally processed, partially treated or raw material intended for human consumption and including beverages, chewing gum and all substances used in the manufacture, preparation and processing of foods, excluding cosmetics or tobacco or substances used solely as medicaments” (*Décret* *n^o^* *06-259/P-RM du 3 juin 2006*, 2)
Mauritius	“…(a) means any article or substance meant for human consumption and includes (i) drinks and bottled water; (ii) chewing gum and other products of similar nature and use; and (iii) articles and substances used or intended for use as ingredients in the composition or preparation of food; (b) does not include (i) live animals, birds or live fish which are not used for human consumption while they are alive; (ii) fodder or feeding stuffs for animals, birds or fish; (iii) drugs or medicine as defined in the Pharmacy Act; and (iv) hormonal products or veterinary products for use in livestock feed” (*Food Act 1998*, 2.)
Morocco	“…any plant or animal product, raw or wholly or partly processed, intended for human consumption including beverages, gum and all products that have been used for the production and preparation or processing of food. This term does not cover plants before harvest and live animals, with the exception of those prepared for human consumption, as they are, such as shellfish, and do not cover medicines, cosmetics and tobacco” (*Loi n^o^ 28-7*, Art. 2.1)
Mozambique	“…any substance which is consumed in the natural state, semi-prepared or processed, intended for human consumption, including beverages, chewing gum and any other substance used in their manufacture, preparation or treatment. Cosmetics, tobacco and substances used solely as medicine are excluded.” (*Decreto n^o^ 15/2006 de 22 de Juhno*, Art. 1 g))
Namibia	“…means any article or substance (except a medicine as defined in the Medicines and Related Substances Control Act, 1965 [Act 10l of 1965]) ordinarily eaten or drunk by man, or purporting to be suitable, or manufactured or sold, for human consumption, and includes any part or ingredient of any such article or substance, or any substance used or intended or destined to be used as a part or ingredient of any such article or substance” (*Foodstuffs, Cosmetics and Disinfectants Ordinance 18 of 1979*, 1)
Niger	“…treated, partially treated or unprocessed substance intended for human consumption and including beverages, chewing gums and all substances used in the manufacture, preparation and processing of foods excluding substances used solely in the form of medicines, cosmetics or tobacco” (*Décret n^o^ 2011-616/PRN/MEL*, Art. 7/3)
Nigeria	“…includes any article manufactured, processed, packaged, sold or advertised for use as food or drink for human consumption, chewing gum and any ingredient which may be mixed with food for any purpose whatsoever and excludes (a) live animals, birds or fish; (b) articles or substances used as drugs” (*Food and Drugs (Amendment) Decree 1999*, 7)
Rwanda	“…any items other than pharmaceutical products, cosmetics and tobacco used as food or drink for human beings and include any substance used in the manufacture or treatment of food” (*Itegeko n^o^ 47/2012 ryo kuwa 14/01/2013*, Art. 2)
Senegal	“…any treated, partially treated or unwrought substance intended for human consumption, and includes beverages, chewing gum and all substances used in the manufacture, preparation and processing of foods, excluding substances used only in form of drugs, cosmetics or tobacco” (*Règlement n^o^ 007/2007/CM/UEMOA*, 1)
Seychelles	“…means any substance, whether processed, semi-processed or raw, which is prepared, sold, represented or intended for human consumption, and includes drinks, bottled and packaged water, chewing gum, other products of similar nature or use and any article, substance or ingredients used in the composition, manufacture, preparation or treatment of food but does not include (a) cosmetics; (b) tobacco; (c) plants prior to harvesting; (d) live animals, birds or live fish which are not used for human consumption while they are alive, (excluding shellfish), unless they are prepared for placing on the market for human consumption; (e) fodder or feed for animals, birds or fish; (f) drugs or medicinal products; (g) hormonal products or veterinary products for use in livestock feed; and (h) residues and contaminants” (*Food Act, 2014*, Art. 2)
Sierra Leone	“…means any article used as food or drink for human consumption, other than drugs or water, and includes (a) any article which is intended for use in the composition or preparation of food; and (b) any flavouring matter or condiment and (c) any colouring matter intended for use in food” (*Act №23*, 2)
Somalia	“…animal products such as meat, milk, eggs, honey, oil, bones, skin, etc” ** *(Veterinary Law Code*, Section 1)
South Africa	“…any article or substance (except a medicine defined in Medicine and Related Substances Act (Act 101 of 1965)) ordinarily eaten or drunk by a person or purporting to be suitable, or manufactured or sold, for human consumption, and includes any part or ingredient of any such article or substance, or any substance used or intended or destined to be used as a part or ingredient of any such article or substance” (*Foodstuffs, Cosmetic, and Disinfectants Act №54 of 1972*, 1)
Sudan	“…means foods or beverages that are prepared or distributed are intended for use in human consumption and include any other substances that are part of their manufacture or part of these substances, including also dairy products.” (*قانون رقم ٥٦٩ لسنة ١٩٧٣م لرقابة الأطعمة*, 2)
Tanzania	“…means any article other than drugs, cosmetics and tobacco used as food or drink for human consumption and includes any substance used in manufacture or treatment of food” (*The Tanzania Food, Drugs, and Cosmetics Act*, Art. 3)
Tunisia	“…all material or transformed or untransformed product that is intended for eating or chewing or adapted to eating or chewing by man” (*قانون عدد 25 لسنة 2019*, 4.1)
Uganda	“…includes drink, chewing gum and other products of a like nature and use, and articles and substances used as ingredients in the preparation of food or drink or of such products, but does not include (i) water, live animals or birds; (ii) fodder or feeding stuffs for animals, birds or fish; or (iii) articles or substances used only as drugs” (*The Food and Drugs Act*, 1)
Zambia	“…includes any article manufactured, sold or represented for use as food or drink for human consumption, chewing gum, and any ingredient of such food, drink or chewing gum” (*The Food and Drugs Act*, 2)
Zimbabwe	“…means any substance which is, in whole or in part, intended for human consumption or which is intended for entry into, or to be used in the manufacture of, any such substance” (*Food and Food Standards Act*, 3)

* This part must have been amended since it is no longer part of this Act, ** definition of ‘animal product’.

**Table 5 foods-09-00502-t005:** Summary of the references towards edible insects within African wildlife management legislation; numbers refer to the specific article of the regulation in question; for Cape Verde, Namibia, Seychelles, Somalia, South Sudan, and Zambia, regulations’ definitions were not applicable to insects; no applicable regulations were found for Libya, São Tomé and Príncipe, Senegal, Sudan, and Zimbabwe.

Country	Regulation	Insects Included	Insects a Usable Resource	Insects Used as Foodstuff	Insect Used Privately	Insects Used Commercially	Additional Information
Algeria	*Loi n^o^ 14/07*	2	2	2	2	2	permission needed (Art. 5); impact study requested (Art. 8); payment for rights
Angola	*Lei n^o^ 6/17*	4.45	4.65	12.1	65, 98	4.43	Title II: sustainable management of forests, Title III: that of animals
Benin	*Loi n^o^ 2002-16*	4	36, 45	96	97	75, 102	Art. 34 states that non-endangered species still benefit from general animal protection measures; permit needed (Art. 73); Chapter III on wildlife rearing
Botswana	*Wildlife Conservation and National Parks Act*	2	19	19	19	39	Special game licenses for veld produce gatherers (30); wildlife management areas (Part III)
	*Agricultural Resources Conservation (Utilization of Veld Products) Regulations*	s *	2	2	3	4	Applies to phane caterpillars (*Gonimbrasia belina*) being a “veld product”; permit if harvest exceeds 10 kg/person/month
Burkina Faso	*Décret n^o^ 96-061/PRES/PM/MEE/MATS/ MEFP/MCIA/MTT*	?	III	III	21	27, 29	No definition of “animal” provided; mammals and birds: permits depending on the level of catching and the species
	*Loi n ^o^ 003-2011/AN*	107	115	115	115	115	Title III on the wildlife usage incl. farming
Burundi	*Loi n^o^ 1/07*	4.25, 4.26	144	144	144	144	Chapter IV, Section 3 deals with “non-wood forest products”
Cameroon	*Loi n^o^ 94/01*	3	8	8	8	101	Local populations harvesting forest products for personal use (Art. 8), permission is needed (Art. 99)
Central African Republic	*Loi n^o^ 08.022*	65	65	65	65	66–75	Only industrial usage of “forest products others than wood” is addressed
Chad	*Loi n^o^ 14/PR/2008*	I.2, II.2, III. 95	II.2, III. 141	III. 76	III. 72–77, III. 143	III. 78–88, III. 154, III. 179	Definition of “forest product” is prone to include insects; Extent of usage depends on the forest type; traditional usages is free and does not require a permit when used for personal needs (III.76); commercial use is subject to taxes and can be done by the owners; special section for ranched fauna (III. 180–189).
Comoros	*Loi-cadre n^o^ 94-018*	40	40	40	40	40	All wildlife usage (41) and introduction of foreign species (44) need an official permit
Congo, Democratic Republic (Kinshasa)	*Loi n^o^ 82-0022*	2					“game animals” are part of the term “fauna”, which comprises all vertebrate and invertebrate species
	*Arrêté n^o^ 014/CAB/MIN/ENV/2004*		33			38	Taxation and basic requirements for keeping wild animals
Congo, Republic (Brazzaville)	*Loi n^o^ 48/83 du 21/04/1983*	2					Insects classified as in no need of special protection; all hunting requires a permit (Article 7); law does not refer to private or commercial uses.
Djibouti	*Décret n^o^ 2004-0065/PR/MHUEAT*	2	2	2	2	2	Insects included in “biodiversity”; no wild animals may be managed.
Egypt	*Law No. 102 of 1983*	2	2	2	2	2	Applies to protected areas only
Equatorial Guinea	*Ley n.^o^ 8/1.988*	2	3	3	46	49–63	Permit needed for hunting (Article 9)
Eritrea	*Proclamation № 155/2006*	2	25	25	25, 29, 30	25, 30	Permit needed for hunting (Article 25); specifications in case hunting is done due to necessity (29) and for wildlife farming (30)
Eswatini	*Environment Management Act, 2002*	2(1)					Insects part of “natural resources”, attended by the government
Ethiopia	*Proclamation № 541/2007*	2	2	2, 8	8	8, 12	Permit needed for hunting (Article 8) and trading (12)
Gabon	*Loi n^o^ 016/01*	4	163	199	199, 252–261	177, 233–243	Permit needed for any kind of wildlife-related activity
	*Décret n^o^ 692/PR/MEFEPEPN*	2	2	2	7	7	Subsidiary hunting is allowed and free, but products may be sold only within the community.
Gambia	*National Environment Management Act (NEMA)*	2					Insects are part of “biological diversity”, but no hints to a (sustainable) use are provided
	*Biodiversity and Wildlife Act, 2003*	2	VI	VI	VI	VIII	Same definition as in NEMA; harvesting license mandatory; commercial use of biological resources prohibited
Ghana	*Wild Animals Preservation Act, 1961*	12					Some edible species would be in the so-called “5th schedule” which allows one to take large amounts of animals to reduce their numbers.
	*Wildlife Conservation Regulations*						Theoretically, insects would enter the “4th schedule” in which an application has to be written to the authorities.
Guinea	*Loi U97/038/An du 9 Decembre*	2	61	61	61	62	Insects would be wild fauna, although other (vertebrate) taxa are mentioned in particular; no commercialization allowed.
Guinea-Bissau	*Decreto-Lei n^o^ 5/2011*	2	15	15	15	24	Insects part of “forest resources”
Ivory Coast	*Loi n^o^ 65-255 &94-442*						Insects are beyond the animal definition
Kenya	*Wildlife Conservation and Management Act, 2013, No. 47 of 2013*	3	81	81	81	8th s *	Permits required; game farming allowed for butterflies (10th schedule),
Lesotho	*Historical Monuments, Relics, Fauna and Flora Act, Act 41 of 1967*	2					Among insects, only refers to bees; any activity requires written consent by the authorities
Liberia	*New Wildlife and National Parks Act*	1	27	27	27	28	Keeping wild animals only with a permit
Madagascar	*Ordonnance n^o^ 60-126 du 3 Octobre 1960*	1	3	3	3	3	Insects may fit into the category “vermin birds and other animals” and may be harvested indiscreetly
Malawi	*National Parks and Wildlife Act, № 11 of 1992*	2	45	45	69	Part X	Insects seem to fall into the unprotected animal category
Mali	*Loi n^o^ 2018-036/ du 27 Juin 2018 fixant les principes de la gestion de la faune et de son habitat*	2	78	166	143	113, 143	Insects are part of fauna; harvesting permit necessary; rearing of any kind of wild animal species in confinement is allowed (Chapter X)
Mauritania	*Loi n^o^ 97-006 abrogeant et remplaçant la loi n° 75-003 du 15 janvier portant la chasse et de la protection de la nature*	7	7	7	7	20	Animal gathering is termed hunting and is regulated, but insects are not mentioned explicitly, and fauna is divided in completely and partially protected vertebrates
Mauritius	*Native Terrestrial Biodiversity and National Parks Act 2015*	2	26	26	26	26	Insect gathering may be seen as a kind of hunting and thus requires a permit
Morocco	*Loi n^o^ 29-05 relative à la protection des espèces de flore et de faune sauvages et au contrôle de leur commerce*	2					Definition in Art. 2 include insects, but the hunting regulation refers to endangered (vertebrate) species only; other hunting regulations do not refer to species.
Mozambique	*Lei n^o^ 10/99*	1	20	20	20	23	Permits are required
Niger	*Loi n^o^ 98-07 du 29 avril 1998 fixant le Régime de la Chasse et de la Protection de la Faune*	4	5	5	5	18	Permits requested for hunting and selling
Nigeria	*Forest Regulations*	2	20	20	20	20	Insects are “minor forest produce” as defined by Art. 2 Forest Law; harvesting them requires a permit
	*Wild Animals Law*	2	43	43	43		Killing animals in times of famine
Rwanda	*Itegeko n^o^ 70/2013 ryokuwa 07/09/2013 rigenga urosobe rw’ibinyabuzima mu Rwanda*	2	29	29	29		Using biodiversity requires a permit
Sierra Leone	*Wildlife Conservation Act, 1972*	2	37	37	37	37	Permits required for hunting animals not contemplated in the schedules (e.g., insects)
South Africa	*National Forests Act*	2	22	22	22	28	Regulates the use and selling of “forest produces” in state forests
Tanzania	*Wildlife Conservation Act, 2008*	3	55	55	55	55, 89	Permits and licenses required for any wildlife-related activity
Togo	*Loi n^o^ 2008-09 portant Code Forestier*	2	Title IV	Title IV	Title IV	14, Title IV	Permits and licenses required for any wildlife-related activity
Tunisia	*Loi n^o^ 20 portant Code forestier*	3	215	215	211, 215		Animal products from forests may not be placed on the market
Uganda	*Uganda Wildlife Statute, 1996*	2	Part VI	Part VI	Part VI	Part VI, part VII	Permits and licenses required for any wildlife-related activity

* s = schedule.

**Table 6 foods-09-00502-t006:** Pest insect legislation pertaining edible insect species (contained in “all pest insect species”); No data could be obtained for Angola, Djibouti, Equatorial Guinea, Ethiopia, Gabon, Lesotho, Senegal, Sierra Leone, Somalia, South Sudan, Sudan, and Togo.

Country	Law
Algeria	*Loi n^o^ 87-17*
Benin	*Loi n^o^ 91-001, Décret n^o^ 92-258*
Botswana	*Plant Protection Act 2007*
Burkina Faso	*Loi n^o^ 025-2017/AN*
Burundi	*Décret-loi n^o^ 1/033*
Cameroon	*Loi n^o^ 2003/003*
Cape Verde	*Lei n^o^ 29/VIII/2013.*
Central African Republic	*Loi n^o^ 62-350*
Chad	*Loi n^o^ 14/PR/95*
Comoros	*Décret du 24 juin 1903*
Congo-Kinshasa	*Décret n^o^ 05/162*
Congo-Brazzaville	*Loi n^o^ 52-1256*
Egypt	*قانون الزراعة رقم 53 لسنة 1966*
Eritrea	*Plant Quarantine Proclamation*
Eswatini	*Plant Control Act, 1981*
Gambia	*Plant Importation and Regulation Act.*
Ghana	*Plants and Fertilizer Act, 2010*
Guinea	*Loi L/92/027/CTRN*
Guinea-Bissau	*Decreto-Lei NQ 4/99*
Ivory Coast	*Loi n^o^ 64-490*
Kenya	*Plant Protection Act*
Liberia	*Agricultural Law*
Libya	*قانون رقم 27 لسنة 1968 بشأن وقاية النباتات*
Madagascar	*Ordonnance n^o^ 86-013*
Malawi	*Plant Protection Act, 1969*
Mali	*Loi n^o^ 02-013 du 03 juin 2002*
Mauritania	*Loi n^o^ 2000-042*
Mauritius	*Plant Protection Act 2006*
Morocco	*Dahir du 20 septembre 1927*
Mozambique	*Decreto n^o^ 5/2009 de 1 de Junho*
Namibia	*Plant Quarantine Act 2008*
Niger	*Loi n^o^ 2015-35 du 26 mai 2015*
Nigeria	*Agriculture (Control of Importation) Act*
Rwanda	*Itegeko n^o^ 16/2016*
São Tomé & Príncipe	*Decreto Lei n^o^ 5/2016*
Seychelles	*Plant Protection Act*
South Africa	*Agricultural Pests Act*
Tanzania	*Plant Protection Act*
Tunisia	*Loi n^o^ 92-72*
Uganda	*Plant Protection and Health Act*
Zambia	*Plant Pests and Diseases Act*
Zimbabwe	*Plant Pests and Diseases Act*

**Table 7 foods-09-00502-t007:** Pest insect legislation pertaining edible insect species (as presented in [11]): locusts (Orthoptera; all species are Acrididae, except for *Zonocerus variegatus* (Zygomorphidae)).

Species	Country	Law
Acrididae spp. (“locusts”)	Botswana	*Locusts Act*
	Burundi	*Ordonnance n^o^ 53*
	Chad	*Loi n^o^ 005/PR/2007*
	Egypt	*قرار وزاري رقم ١٨ لسنة ١٩٦٧*
	Keyna	*Plant Protection Rules*
	Lesotho	*Locust Destruction Proclamation*
	Madagascar	*Décret n^o^ 2017-064*
	Mauritania	*Décret n^o^ 2002-062 du 25 juillet 2002*
	Namibia	*Regulations relating to the Destroying of Locusts*
	Sudan	*قانون أبادة الجراد لسنة 1907م*
*Anacridium melanorhodon*	Mauritius	*Plant Protection Act 2006*
	Seychelles	*Plant Protection Act*
*Locusta migratoria migratorioides*	Kenya	*Plant Protection Order, Plant Protection Rules*
	Zambia	*Plant Pests and Diseases (Pests and Alternate Hosts) Order*
	Zimbabwe	*Locust Control Act [Chapter 19:06].*
*Locustana pardalina*	Eswatini ***	*Plant Control Act, 1981*
	Zimbabwe	*Locust Control Act*
*Melanoplus differentialis*	Mauritius	*Plant Protection Act 2006*
*Nomadacris septemfasciata*	Eswatini	*Plant Control Act, 1981*
	Kenya	*Plant Protection Order, Plant Protection Rules*
	Zambia	*Plant Pests and Diseases (Pests and Alternate Hosts) Order*
	Zimbabwe	*Locust Control Act [Chapter 19:06].*
*Schistocerca gregaria*	Egypt	*قرار وزاري رقم ١٧ لسنة ١٩٦٧ بوضع نظام مكافحة الجراد الصحراوي*
	Kenya	*Plant Protection Order, Plant Protection Rules*
	Zimbabwe	*Locust Control Act*
*Zonocerus variegatus*	Mozambique	*Decreto n^o^ 5/2009*

* termed “Locusta pardelina” in this act.

**Table 8 foods-09-00502-t008:** Pest insect legislation pertaining edible insect species (as presented in [11]: butterflies and moths (Lepidoptera).

Family	Species	Country	Law
Cossidae	*Cossus cossus*	Algeria	*Décret exécutif n^o^ 95-387*
Crambidae	*Chilo* spp.	Kenya	*Plant Protection Order*
		Mozambique	*Decreto n^o^ 5/2009*
	*Ostrinia furnacalis*	Botswana	*Plant Protection Act 2007*
Gelechiidae	*Pectinophora gossypiella*	Zambia	*Plant Pests and Diseases (Pest Control) Regulations, Plant Pests and Diseases (Pests and Alternate Hosts) Order*
Noctuidae	*Busseola fusca*	Kenya	*Plant Protection Order*
	*Helicoverpa* spp.	Cape Verde	*Portaria n^o^ 37/2015*
	*Heliothis* spp.	Botswana	*Plant Protection Act 2007*
		Seychelles	*Plant Protection Act*
	*H. zea*	Tunisia	*Arrêté du Ministre de l’agriculture du 31 mai 2012*
	*Spodoptera spp.*	Botswana	*Plant Protection Act 2007*
		Mozambique	*Decreto n^o^ 5/2009*
		Tunisia	*Arrêté du Ministre de l’agriculture du 31 mai 2012*
	*S. frugiperda*	Mozambique	*Decreto n^o^ 5/2009*
		South Africa	*Control Measures Relating to Fall Armyworm*
		Tunisia	*Arrêté du Ministre de l’agriculture du 31 mai 2012*
Pieridae	*Pieris* spp.	Mozambique	*Decreto n^o^ 5/2009*
Pyralidae	*Chilo* spp.	Benin	*Arrêté interministériel n^o^ 128*
		Botswana	*Plant Protection Act 2007*
Sphingidae	*Acherontia styx*	Mozambique	*Decreto n^o^ 5/2009*
	*Agrius convolvuli*	Botswana	*Plant Protection Act 2007*
	*Clanis bilineata*	Mozambique	*Decreto n^o^ 5/2009*
	*Erinnyis ello*	Mozambique	*Decreto n^o^ 5/2009*

**Table 9 foods-09-00502-t009:** Pest insect legislation pertaining edible insect species (as presented in [11]): beetles (Coleoptera); Eswatini also refers to “woodborer beetles”, but without any species information, a term that, in fact, can be used to several taxa.

Family	Species	Country	Law
Ceram-bycidae	*Anoplophora* spp.	Tunisia	*Arrêté du Ministre de l’agriculture du 31 mai 2012*
	*A. chinensis*	Mauritius	*Plant Protection Act 2006*
	*A. glabripennis*	Mauritius	*Plant Protection Act 2006*
	*Monochamus* spp.	Tunisia	*Arrêté du Ministre de l’agriculture du 31 mai 2012*
Chryso-melidae	*Leptinotarsa* spp.	Benin	*Arrêté interministériel n^o^ 128*
	*L. decemlineata*	Malawi	*Plant Protection Act, 1969*
		Mauritania	*Arrêté n^o^ R-0031257 du 12 novembre 2002*
		Mauritius	*Plant Protection Act 2006*
		Mozambique	*Decreto n^o^ 5/2009*
		Seychelles	*Plant Protection Act*
		Tunisia	*Arrêté du Ministre de l’agriculture du 31 mai 2012*
		Zambia	*Plant Pests and Diseases (Pest Control) Regulations, Plant Pests and Diseases (Pests and Alternate Hosts) Order*
Curculi-onidae	*Anthonomus* spp.	Mozambique	*Decreto n^o^ 5/2009*
	*Cosmopolites sordidus*	Kenya	*Plant Protection Order*
		Mauritania	*Arrêté n^o^ R-0031257 du 12 novembre 2002*
	*Rhynchophorus* spp.	Benin	*Arrêté interministériel n^o^ 128*
		Mauritania	*Arrêté n^o^ R-0031257 du 12 novembre 2002*
	*R. ferrugineus*	Mauritius	*Plant Protection Act 2006*
		Morocco	*Arrêté conjoint (…) n^o^ 287-09*
		Tunisia	*Arrêté du Ministre de l’agriculture du 31 mai 2012*
	*R. phoenicis*	Mauritius	*Plant Protection Act 2006*
	*Scyphophorus acupunctatus*	Kenya	*Plant Protection Order*
		Mozambique	*Decreto n^o^ 5/2009*
	*Sitophilus oryzae*	Kenya	*Plant Protection Order*
		Zambia	*Plant Pests and Diseases (Pests and Alternate Hosts) Order*
Scarabeidae	*Adoretus* spp.	Botswana	*Plant Protection Act 2007*
		Benin	*Arrêté interministériel n^o^ 128*
	*Amphamalion* spp.	Botswana	*Plant Protection Act 2007*
	*Anomala* spp.	Botswana	*Plant Protection Act 2007*
	*Leucophora* spp.	Mozambique	*Decreto n^o^ 5/2009*
	*Oryctes* spp.	Benin	*Arrêté interministériel n^o^ 128*
	*Popilia* spp.	Benin	*Arrêté interministériel n^o^ 128*
	*P. japonica*	Malawi	*Plant Protection Act, 1969*
		Tunisia	*Arrêté du Ministre de l’agriculture du 31 mai 2012*
		Zambia	*Plant Pests and Diseases (Pest Control) Regulations*
	*Xylotrupes gideon*	Botswana	*Plant Protection Act 2007*
Tenebrionidae	*Alphitobius* spp.	Kenya	*Plant Protection Order*
	*A. diaperinus*	Zambia	*Plant Pests and Diseases (Pests and Alternate Hosts) Order*
	*A. levigatus*	Mozambique	*Decreto n^o^ 5/2009*
		Zambia	*Plant Pests and Diseases (Pests and Alternate Hosts) Order*
	*Tenebrio* spp.	Kenya	*Plant Protection Order*
		Zambia	*Plant Pests and Diseases (Pests and Alternate Hosts) Order*
	*T. molitor*	Kenya	*Plant Protection Order*
		Zambia	*Plant Pests and Diseases (Pests and Alternate Hosts) Order*
	*Tribolium castaneum*	Kenya	*Plant Protection Order*
		Zambia	*Plant Pests and Diseases (Pests and Alternate Hosts) Order*
	*T. confusum*	Kenya	*Plant Protection Order*
		Zambia	*Plant Pests and Diseases (Pests and Alternate Hosts) Order*

**Table 10 foods-09-00502-t010:** Pest insect legislation pertaining edible insect species (as presented in [11]): other orders.

Order	Family	Species	Country	Law
Blattodea	Blattidae	*Blatta orientalis*	Zambia	*Plant Pests and Diseases (Pests and Alternate Hosts) Order*
		*Periplaneta americana*	Zambia	*Plant Pests and Diseases (Pests and Alternate Hosts) Order*
	Ectobiidae	*Blatella germanica*	Zambia	*Plant Pests and Diseases (Pests and Alternate Hosts) Order*
Diptera	Tephritidae	*Anastrepha* spp.	Benin	*Arrêté interministériel n^o^ 128*
			Botswana	*Plant Protection Act 2007*
			Mauritania	*Arrêté n^o^ R-0031257 du 12 novembre 2002*
			Mozambique	*Decreto n^o^ 5/2009*
		*A. ludens*	Mauritius	*Plant Protection Act 2006*
			Tunisia	*Arrêté du Ministre de l’agriculture du 31 mai 2012*
	Tipulidae	*Tipula paludosa*	Botswana	*Plant Protection Act 2007*
Hemiptera	Alydidae	*Leptocorisa acuta*	Mozambique	*Decreto n^o^ 5/2009*
		*L. oratorius*	Mozambique	*Decreto n^o^ 5/2009*
	Psyllidae	*Psylla* spp.	Mozambique	*Decreto n^o^ 5/2009*
	Pyrrhocoridae	*Dysdercus* spp.	Botswana	*Plant Protection Act 2007*
	Tessaratomidae	*Tessaratoma papillosa*	Mozambique	*Decreto n^o^ 5/2009*
Hymenoptera	Formicidae	*Atta cephalotes*	Botswana	*Plant Protection Act 2007*

**Table 11 foods-09-00502-t011:** Summary of the legal context of edible insects in Africa.

Country	Native Edible Species	Edible Insects Tolerated	Suitable Foodstuff Definition	Wildlife Management—Private	Wildlife Management—Commercial	General Pest Management	Edible Specified as Pests
Algeria	(−)	+	+	+	+	+	+
Angola	+		+	+	+		
Benin	+	+	+	+	+	+	+
Botswana	+	+		+	+	+	+
Burkina Faso	+	+	+	−	−	+	
Burundi	+	+		+	+	+	+
Cameroon	+	+		+	−	+	
Cape Verde	−	(−)	+	−	−	+	+
Central African Republic	+	+	+	+	+	+	
Chad	+			+	+	+	+
Comoros	−			+	+	+	
Congo, Democratic Republic (Kinshasa)	+	+		+	+	+	
Congo, Republic (Brazzaville)	+			+	+	+	
Djibouti	−			−	−		
Egypt	+		+	+	+	+	+
Equatorial Guinea	+			+	+		
Eritrea	−			+	+	+	
Eswatini	+		+	+	+	+	+
Ethiopia	+		+	+	+		
Gabon	+			+	+		
Gambia	+			+	−	+	
Ghana	+		+	+		+	
Guinea	+	+		+	−	+	
Guinea-Bissau	+		+	+	+	+	
Ivory Coast	+		+	−	−	+	
Kenya	+		+	+	+	+	+
Lesotho	+			+	+		+
Liberia	+		+	+	+	+	
Libya	+					+	
Madagascar	+	+	+	+	+	+	+
Malawi	+		+	+	+	+	+
Mali	+		+	+	+	+	
Mauritania	(−)		+	+	+	+	+
Mauritius	+		+	+	+	+	
Morocco	+	+	+			+	+
Mozambique	+		+	+	+	+	+
Namibia	+		+	−	−	+	+
Niger	+	+	+	+	+	+	
Nigeria	+		+	+	+	+	
Rwanda	(−)		+	+	+	+	
São Tomé and Príncipe	+					+	
Senegal	+		+				
Seychelles	−		+	−	−	+	+
Sierra Leone	+		+	+	+		
Somalia	(−)		+	−	−		
South Africa	+	+	+	+	+	+	+
South Sudan	+			−	−		
Sudan	+		+				+
Tanzania	+	+	+	+	+	+	
Togo	+	+		+	+	.	
Tunisia	(−)	+	+	+	−	+	+
Uganda	+		+	+	+	+	
Zambia	+		+	−	−	+	+
Zimbabwe	+		+			+	+

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
