# Peer review of "Edible Insects in Africa in Terms of Food, Wildlife Resource, and Pest Management Legislation"

_foods, 2020, doi:10.3390/foods9040502_

Round 1

Reviewer 1 Report

Dear Authors,

It was a plasure to revise your important contribution

  • the paper addresses a very important issue (legal status of insects harvesting, production and consumption in African countries) and attempts to analyse the current drawbacks around incorporating edible insects in a clear food safety legal framework.
  • the proposed method includes comprehensive discussion and analysis of insects with references to food law, wildlife resources management and pest management. In this regards the paper appears well structured and includes novel information that certainly required considerable collaborative efforts considering the language barriers existing among African countries and likely obstacles related to the availability of well references and trusted sources.
  • presentation and discussion are in line with the proposed aims and methods
  • in regards to the current EU legal framework, I suggest to mention from te beginning in the paragraph 'The European Union as a starting point for legal considerations', the Novel Food Regulation (REG [EC] 2015/2283) that finally assigns to edible insects a clearer legal status than before 2018. This legislative step is remarkable because tries to overcome the different national legislation andd to provide EU insect producers, suppliers and sellers with a clearer regulatory environment to plan investment and conduct marketing activities, and of course consumers with safer food products. 
  • to provide the readers with a more updated EU legal perspective, I would suggest mentioning the recent Commission regulation proposal, which modifies Annex III of Regulation 853/2004, and provides: - the definition of insects for human consumption; - establishes the requirements for the substrates used for breeding and the procedure for placing them on the market. 

Recommendations

It is advisalbe to elaborate more in the tight conclusions, that seems to be somewhat disproportionate compared to the extensive article text.

Based on the identified gaps and constraints, mainly referring to the lack of clear status of edible insect in the Africa states's food laws, I invite the auhors to formulate very general recommendations for African stakeholders, eg. producers, legal institutions, policymakers...

THe English language is understandable, no needed proof reading.

Author Response

Reviewer1 Comments
Dear Authors, It was a plasure to revise your important contribution
  • the paper addresses a very important issue (legal status of insects harvesting, production and consumption in African countries) and attempts to analyse the current drawbacks around incorporating edible insects in a clear food safety legal framework.
  • the proposed method includes comprehensive discussion and analysis of insects with references to food law, wildlife resources management and pest management. In this regards the paper appears well structured and includes novel information that certainly required considerable collaborative efforts considering the language barriers existing among African countries and likely obstacles related to the availability of well references and trusted sources.
  • presentation and discussion are in line with the proposed aims and methods
Thank you very much.
  • in regards to the current EU legal framework, I suggest to mention from te beginning in the paragraph 'The European Union as a starting point for legal considerations', the Novel Food Regulation (REG [EC] 2015/2283) that finally assigns to edible insects a clearer legal status than before 2018. This legislative step is remarkable because tries to overcome the different national legislation andd to provide EU insect producers, suppliers and sellers with a clearer regulatory environment to plan investment and conduct marketing activities, and of course consumers with safer food products. 
We initially thought of including more detail on the EU framework, but opted against it for reasons of space. However, all reviewers requested it, and so we included a small section.
  • to provide the readers with a more updated EU legal perspective, I would suggest mentioning the recent Commission regulation proposal, which modifies Annex III of Regulation 853/2004, and provides: - the definition of insects for human consumption; - establishes the requirements for the substrates used for breeding and the procedure for placing them on the market. 
I know of this proposal, but it has not been ratified so far. We included the data with a reference to the draft status.
Recommendations ·         It is advisalbe to elaborate more in the tight conclusions, that seems to be somewhat disproportionate compared to the extensive article text. ·         Based on the identified gaps and constraints, mainly referring to the lack of clear status of edible insect in the Africa states's food laws, I invite the auhors to formulate very general recommendations for African stakeholders, eg. producers, legal institutions, policymakers... That is an interesting comment. We would not have dared to go that far but we include a set of recommendations from our point of view.
·         THe English language is understandable, no needed proof reading. Thanks for the compliment. None of us authors is a natural-born native speaker.

Reviewer 2 Report

1)    The manuscript is very interesting and useful to the scientific community, although it is very long

2)    The manuscript is an extensive examination of food law in Africa, focusing on legal status of edible insects

for this reason it’s my opinion that the authors should modify the title to give a precise idea about the content of the manuscript

3)    I suggest that authors explicitly mention the Novel Food Regulation (REG [EC] 2015/2283), especially in the discussion section of the manuscript

4)     In the conclusions section, I suggest to the authors to briefly insert proposals for the regulation of edible insects in Africa

     Line 400: …Faso and….

     Line 421: Cabo Verde or  Cape Verde?      always use the same name

      Tab 5:  „natural resources“,  “      “

Author Response

Reviewer2 Comment
1)    The manuscript is very interesting and useful to the scientific community, although it is very long Thank you. The length is due to the amount of countries to be considered, as you know.
2)    The manuscript is an extensive examination of food law in Africa, focusing on legal status of edible insects for this reason it’s my opinion that the authors should modify the title to give a precise idea about the content of the manuscript We agree to change the title. However, food law is only one section of many that pertain using edible insects, so also included these other sections.
3)    I suggest that authors explicitly mention the Novel Food Regulation (REG [EC] 2015/2283), especially in the discussion section of the manuscript We initially thought of including more detail on the EU framework, but opted against it for reasons of space. However, all reviewers requested it, and so we included a small section in the introduction. We also mentioned the novel food regulation in the discussion
4)     In the conclusions section, I suggest to the authors to briefly insert proposals for the regulation of edible insects in Africa That is an interesting comment. We would not have dared to go that far but we include a set of recommendations from our point of view.
Line 400: …Faso and…. Done, thank you
     Line 421: Cabo Verde or  Cape Verde?      always use the same name According to Wikipedia, both versions are correct. However, we understand the need of uniformity and changed “Cabo” to “Cape” throughout the manuscript.
      Tab 5:  „natural resources“,  “      “ I must admit I don’t find the mistake. The only row containing “natural resources” is the Eswatini one, and everything seems to be okay. Please be more precise on the kind of mistake.

Reviewer 3 Report

The manuscript “Legal status of edible insects in Africa” is a well-written comprehensive review, that, undoubtfully, will contribute to our knowledge and will reveal the current status of edible insects as food in African countries. The authors did a great job on collecting and processing information and I certainly recommend the manuscript for publication after minor corrections.

As a general comment, I would highly recommend improving the Abstract and Introduction. 

Line 23: “they” – clarify

Line 27: add “database” to FAOLEX.

Line 28-32: Not clear. 

Line 35: “Entomophagy” can be added to keywords.

Line 55-58: Add a reference

Line 61: Add “insect” to species

Table 1: Is it possible to provide information if the first 2 insect species were cooked or no (as it has been mentioned for Rhynchophorus phoenicis?

Line 110: in this chapter, Regulation (EU) 2015/2283 on novel foods should be mentioned.

Line 164-165: Check if the space between the paragraphs is required. Follow the same style in the whole manuscript.

Line 362: including insects?

Line 857: Edible insects?

Author Response

Reviewer3 Comment
As a general comment, I would highly recommend improving the Abstract and Introduction.  As for the abstract, we changed the second part of the abstract (originally lines 28 to 32), explaining our findings with more detail, and we included a reference to our recommendations which is a novel element recommended by the other two reviewers. In the introduction, we detailed EU legislation, hoping to have met the reviewer’s expectations.
Line 23: “they” – clarify Thanks for spotting this incongruence. We changed from “they” to “insects”.
Line 27: add “database” to FAOLEX. Text was changed accordingly.
Line 28-32: Not clear.  See first comment, please.
Line 35: “Entomophagy” can be added to keywords. As there is a limit of five keywords, “edible insects” was switched to “entomophagy”
Line 55-58: Add a reference We added Amadi and Kiin-Kabari, 2016.
Line 61: Add “insect” to species Done.
Table 1: Is it possible to provide information if the first 2 insect species were cooked or no (as it has been mentioned for Rhynchophorus phoenicis? This is a good point. We traced the data back and added “raw” in the table.
Line 110: in this chapter, Regulation (EU) 2015/2283 on novel foods should be mentioned. We initially thought of including more detail on the EU framework, but opted against it for reasons of space. However, all reviewers requested it, and so we included a small section.
Line 164-165: Check if the space between the paragraphs is required. Follow the same style in the whole manuscript. This may be a remnant from the time before the first layouting by the publisher. We added spaces when necessary.
Line 362: including insects? This is a good question which we can only answer to the best of our knowledge. Insects do into the foodstuff definition, but are not mentioned explicitly. So I guess it would be possible – we should have to test it directly in Porto Novo or Cotonou and see what happens J. In any way, we added this to the manuscript.
Line 857: Edible insects? Yes, correct, thanks for bringing this to our attention.